# Towards Robust Heterogeneous Graph Explanations under Structural Perturbations

## Abstract

Explaining the prediction process of Graph Neural Networks (GNNs) is critical for enhancing model transparency and trustworthiness. However, real-world graphs are predominantly heterogeneous and often suffer from structural noise, which severely hampers the reliability of existing explanation methods. To address this challenge, we propose RoHeX, a Robust Heterogeneous Graph Neural Network Explainer. RoHeX begins with a theoretical analysis of how different heterogeneous GNN architectures amplify noise through message passing. To mitigate this effect, we introduce a denoising variational inference framework that operates on the graph structure to extract robust latent representations. Furthermore, RoHeX incorporates heterogeneous edge semantics into the subgraph generation process and formulates explanation as an optimization problem under the graph information bottleneck principle. This enables RoHeX to generate explanations that are both semantically meaningful and structurally robust. Extensive experiments on multiple real-world heterogeneous graph datasets demonstrate that RoHeX significantly outperforms state-of-the-art baselines in terms of explanation quality and robustness to noise.

## 1 Introduction

Graph Neural Networks (GNNs) have emerged as powerful tools for learning from graph-structured data, demonstrating strong performance across domains such as social networks [1], citation graphs [2], and recommender systems [3]. Despite these successes, GNNs remain largely opaque: their predictions are difficult to interpret, which limits their deployment in sensitive domains involving fairness, privacy, and security [4, 5].

To address this, GNN explainers aim to reveal the decision rationale behind model predictions by identifying critical substructures. Existing approaches fall into two categories: post-hoc methods [6, 7, 8], which explain a pretrained model without modifying it, and built-in methods [9, 10, 11], which generate explanations during model training. Post-hoc methods are more flexible and generalizable, while built-in methods often yield task-specific insights. However, both struggle under complex, noisy conditions—especially in heterogeneous graphs, which are the norm in real-world settings [12, 13].

Heterogeneous graphs, composed of diverse node and edge types, introduce nontrivial challenges. Their interwoven semantics and irregular structures complicate subgraph extraction. Moreover, real-world graphs commonly exhibit noise such as irrelevant or missing edges, which exacerbate the challenges in achieving robust model explainability [14, 15]. The diversity in distributions, attributes, and application domains across heterogeneous graphs also limits the generalizability of existing explainers. While post-hoc methods offer broader applicability, they remain vulnerable to structural irregularities and noise, which can distort the model's reasoning process. Moreover, noise amplifies structural irregularities and shifts node importance, rendering conventional methods that rely on strict constraints (e.g., subgraph size, connectivity, or budget) ineffective [16].

Submitted to 39th Conference on Neural Information Processing Systems (NeurIPS 2025). Do not distribute.

In this paper, we propose RoHeX, a Robust Heterogeneous Graph Neural Network Explainer. First, we theoretically analyze how noise amplification occurs in heterogeneous GNNs. Second, we introduce a denoising variational inference module that learns robust latent representations by filtering noise in the input graph. Third, we design a heterogeneous explanation generator based on relation-aware attention, which captures the rich semantics across node and edge types. Finally, by integrating the Graph Information Bottleneck (GIB) principle, we reframe explanation as an information-theoretic optimization problem to better handle irregular structures. We validate RoHeX's performance through extensive experiments on multiple real-world datasets, demonstrating its superior ability to handle noise and generate high-quality explanations compared to state-of-the-art GNN explainers.

The contributions of this paper are as follows:

- We present the first systematic analysis of how noise interacts with heterogeneity in GNN explainers, theoretically proving that existing heterogeneous GNNs amplify structural noise and degrade explainability.

- We propose RoHeX, a robust heterogeneous GNN explainer that integrates denoising variational inference and a relation-aware explanation generator, effectively modeling heterogeneity while suppressing noise during explanation generation.

- Extensive experiments on multiple real-world heterogeneous graphs demonstrate that RoHeX consistently outperforms existing explainers in both explanation fidelity and robustness under noisy conditions.

## 2  Problem Definition

### 2.1  Heterogeneous Graph

A heterogeneous graph, denoted as $\mathcal{G} = (\mathbf{A}, \mathbf{X}, \mathcal{A}, \mathcal{R}, \Phi)$, encompasses multiple types of nodes $\mathcal{V}$ and edges $\mathcal{E}$, where $\mathbf{A}$ is the corresponding adjacency matrix, $\mathbf{X}$ represents node features, $\mathcal{A}$ denotes the set of node types, $\mathcal{R}$ signifies the set of edge types, and $\Phi$ represents the set of meta-paths. A meta-path $\phi \in \Phi$ is a path of edges connecting various types of nodes from node $v_1$ to node $v_{l+1}$, such as $\mathcal{A}_1 \xrightarrow{\mathcal{R}_1} \mathcal{A}_2 \xrightarrow{\mathcal{R}_2} \ldots \xrightarrow{\mathcal{R}_l} \mathcal{A}_{l+1}$, where $l$ denotes the length of the meta-path. The label set of $\mathcal{G}$ is denoted as $\mathbf{Y}$, comprising $\mathcal{C}$ categories. Meanwhile, a heterogeneous graph has two mapping functions $\psi(v) : \mathcal{V} \to \mathcal{A}$ and $\varphi(e) : \mathcal{E} \to \mathcal{R}$ that correspond to nodes and edges, respectively.

### 2.2  Heterogeneous Graph Neural Network Explainer

Given a trained GNN model $f$ as the subject of explanation and a heterogeneous graph $\mathcal{G}$, the objective of the GNN explainer is to identify the most influential subgraph $\mathcal{G}_s = (\mathbf{A}_s, \mathbf{X}, \mathcal{A}_s, \mathcal{R}_s)$. Here, $\mathbf{A}_s$ represents the adjacency matrix formed by nodes $\mathcal{V}_s$ and $\mathcal{E}_s$ which significantly contribute to prediction. For the original prediction of GNN model $f$, it can be formalized as follows:

$$\hat{y} = \operatorname*{argmax}_{c \in \mathcal{C}} P_f(c | \mathbf{A}, \mathbf{X}, \mathcal{A}, \mathcal{R}), \tag{1}$$

where $P_f(\cdot)$ represents the prediction function of the GNN model $f$. Current research indicates that graph structural information is crucial for classification tasks [16, 17]. Therefore, our RoHeX focuses on exploring structural noise when generating explanations. The excellent explanation should contain the most critical information to approximate the predicted labels and outcome changing when predicting the remaining part of the original graph, which is as follows:

$$\operatorname*{argmax}_{c \in \mathcal{C}} P_f(c | \mathbf{A}_s, \mathbf{X}, \mathcal{A}_s, \mathcal{R}_s) = \hat{y}. \tag{2}$$

## 3  Methodology

Figure 1 illustrates the overall framework of RoHeX. We begin by applying a denoising variational graph encoder to obtain a robust latent representation of the input graph $\mathcal{G}$. Node embeddings sampled

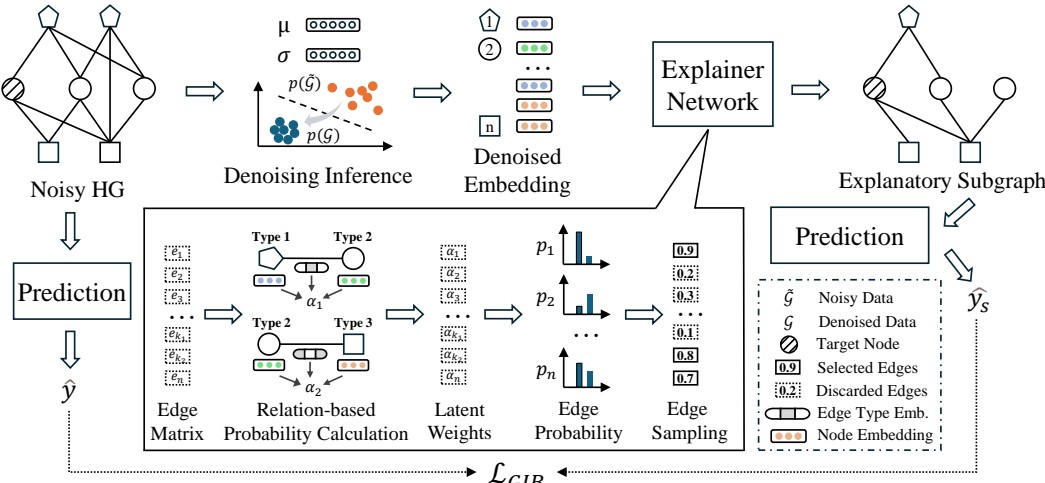

Figure 1: The architecture of our proposed RoHeX. First, the denoised node representations are obtained from the noisy graph via denoising variational inference. Then, the Explainer Network employs the heterogeneous relation-based importance computation method to obtain the weights for different edges. The top $k$ percent of edges are selected as important edges to generate the explanatory subgraph. Finally, the generated explanatory subgraph and the original graph are respectively input into heterogeneous GNN models to obtain predictions, which are used to compute the loss function.

from the latent structural distribution are used to construct edge representations. These edge features are then fed into a heterogeneous relation-aware attention module, which estimates the importance of each edge by modeling the semantics of different edge types. Based on the learned importance scores, RoHeX generates a compact and informative explanatory subgraph. Finally, the entire process is optimized under the graph information bottleneck objective, which adaptively promotes structural sparsity and robustness against irregularities.

## 3.1 Controllable Structural Perturbation for Heterogeneous Graphs

Real-world graph noise often arises from missing or spurious edges and is typically modeled via random edge addition or deletion [18, 19, 20]. Building upon this idea, we introduce a controllable and heterogeneous-aware structural perturbation strategy—a heuristic but flexible method designed to simulate realistic noise, maintain comparability with prior work, and enable targeted evaluation under adversarial or highly irregular settings. Notably, this method can be changed into a graph structure attack method to achieve structural corruption.

For each edge type $r \in \mathcal{R}$, we define its deletion rate $\eta_r^+$ and false addition rate $\eta_r^-$, which represent the probabilities of deleting and adding edges of this type, respectively. The perturbed adjacency $\tilde{\mathbf{A}}_r$ for edge type $r$ is generated as:

$$\tilde{\mathbf{A}}_{ij}^r = \begin{cases} 0 & \text{with probability } \eta_r^+ \cdot \frac{d_i^r + d_j^r}{2\bar{d}^r}, \\ 1 & \text{with probability } \eta_r^- \cdot \frac{1}{d_i^r + d_j^r}, \\ \mathbf{A}_{ij} & \text{otherwise,} \end{cases} \tag{3}$$

where $d_i^r$ is the degree of node $v_i$ under edge type $r$, and $\bar{d}^r = \frac{1}{|\mathcal{V}_r|} \sum_{i \in \mathcal{V}_r} d_i^r$ denotes the average degree for type $r$. This degree-aware perturbation prevents the disproportionate removal of hub-node edges and limits unrealistic edge additions, thus preserving key graph structure properties. To control the overall noise intensity, we introduce a global noise budget $B$, representing the total number of edges to be perturbed. This budget is allocated across edge types according to their importance scores:

$$B_r = B \cdot \frac{s_r}{\sum_{r' \in \mathcal{R}} s_{r'}}, \quad \text{where } s_r = \frac{|\mathcal{E}_r|}{|\mathcal{E}|} \cdot \log |\mathcal{E}_r|, \tag{4}$$

with $s_r$ capturing both the proportion and diversity of edge type $r$. We then calibrate the edge type-specific perturbation rates:

$$\eta_r^+ = \frac{B_r^+}{|\mathcal{E}_r|}, \quad \eta_r^- = \frac{B_r^-}{|\mathcal{V}_r|^2 - |\mathcal{E}_r|}, \quad \text{s.t. } B_r^+ + B_r^- = B_r. \tag{5}$$

Finally, we provide a theoretical analysis showing how degree-aware perturbation impacts node connectivity:

**Theorem 3.1** (Degree-Aware Perturbation). *For node $v_i$, its expected perturbed degree $d_i'$ under the above strategy satisfies:*

$$E[d_i'] = d_i + \sum_{r \in \mathcal{R}} \left( \eta_r^- \cdot \frac{(|\mathcal{V}_r| - d_i^r)}{d_i^r + \bar{d}^r} - \eta_r^+ \cdot \frac{d_i^{r\,2}}{2\bar{d}^r} \right).$$

By incorporating degree balance constraints and noise budget allocation, our method simulates real-world noise more realistically and preserves the structural properties of the original graph.

## 3.2 Noise Analysis and Denoising Variational Inference

We investigate the impact of noise on different approaches for Heterogeneous Graph Neural Network (HGNN). We categorize common HGNN into two classes: meta-path-based and neighborhood aggregation-based methods. Meta-path-based methods typically require defining a meta-path $\phi$, and then capturing information along different relations following the meta-path structure, aggregating this information, such as Paths2Pair [21] and MAGNET [22]. Neighborhood aggregation-based methods simultaneously consider the neighbor node types and edge types and use specific aggregation functions to combine information from different types. Common neighborhood aggregation methods include MHGCN [23] and Simple-HGN [24]. However, these two categories of methods differ in their efficiency of noise propagation [20], and we find that meta-path-based message passing methods amplify the impact of noise.

**Theorem 3.2** (Noise Amplification Effect in HG). *In HG, compared to neighborhood aggregation-based methods, meta-path-based methods can significantly amplify the effect of noisy edges. Specifically, for a node $v_i$ and a newly added noisy edge $e_{ij}$, the factor by which its influence changes is $\frac{d_{v_i} + k}{d_{v_i} + 1}$, where $k$ is the degree of the new neighbor $v_j$ under the noise and $d_{v_i}$ is the degree of $v_i$. When $k > d_{v_i}$, this factor is significantly greater than 1.*

The complete proof of Theorem 3.2 is provided in Appendix D. Based on Theorem 3.2, we employ a neighborhood aggregation method to encode heterogeneous graph and mitigate noise. Given noisy graph data $\tilde{\mathcal{G}}$, our objective is to obtain a denoised version of the standard graph data $\mathcal{G}$. VGAE [25] uses variational inference to derive statistical properties of the graph. The statistical data of latent variables in VGAE can be efficiently inferred from the latent space rather than the observation space, which provides robust graph information. For the standard graph $\mathcal{G}$, it initially generates latent variables $\mathbf{Z}$ from a prior distribution $p(\mathbf{Z})$, such as a Gaussian distribution $\mathcal{N}(\boldsymbol{\mu}, \boldsymbol{\sigma}^2)$. Second, the data $\mathcal{G}$ is generated using a conditional distribution $p(\mathcal{G}|\mathbf{Z})$. VGAE optimizes its parameters by maximizing the likelihood of the observed data, which as follows:

$$\text{KL}(q_\Psi(\mathbf{Z}|\mathcal{G}) \| p_\theta(\mathbf{Z}|\mathcal{G})) + \mathcal{L}(\Psi, \theta; \mathcal{G}), \tag{6}$$

where $\Psi$ is the encoder and $\theta$ represents the parameters to be optimized. Then, the evidence lower bound $\mathcal{L}(\Psi, \theta; \mathcal{G})$ can be expressed as follows:

$$\mathcal{L}(\Psi, \theta; \mathcal{G}) = \mathbb{E}_{q_\Psi(\mathbf{Z}|\mathcal{G})} \left[ \log \frac{p_\theta(\mathbf{Z}, \mathcal{G})}{q_\Psi(\mathbf{Z}|\mathcal{G})} \right] = \mathbb{E}_{q_\Psi(\mathbf{Z}|\mathcal{G})} [\log p_\theta(\mathbf{Z}|\mathcal{G})] - \text{KL} \left( q_\Psi(\mathbf{Z}|\mathcal{G}) \| p(\mathbf{Z}) \right). \tag{7}$$

Variational inference enhances the model's robustness and generalization capabilities [26, 27]. However, due to the differing distributions between noisy heterogeneous graph data and standard graph data, the obtained distribution tends to align with the noisy distribution, potentially misleading the GNN explainer into generating incorrect explanatory subgraphs. Therefore, we introduce a denoising module during the process of variational inference. The original encoder part is modified to:

$$q_\Psi'(\mathbf{Z}|\mathcal{G}) = \int q_\Psi(\mathcal{G}|\tilde{\mathcal{G}}) q(\tilde{\mathcal{G}}|\mathcal{G}) \mathrm{d}\tilde{\mathcal{G}}, \tag{8}$$

where $\Psi$ is the encoder based on $\tilde{\mathcal{G}}$, and $q(\tilde{\mathcal{G}}|\mathcal{G}) = \prod_{r\in\mathcal{R}} q(\tilde{\mathbf{A}}_r|\mathbf{A}_r)$. During this process, the evidence lower bound is expressed as:

$$\mathcal{L}_d = \mathbb{E}_{q'_\Psi(\mathbf{Z}|\mathcal{G})}[\log \frac{p_\theta(\mathbf{Z}, \mathcal{G})}{q'_\Psi(\mathbf{Z}|\mathcal{G})}]. \tag{9}$$

As we need to derive the distribution of the noisy graph data $\tilde{\mathcal{G}}$, this lower bound can be further refined as:

$$\mathcal{L}_d = \mathbb{E}_{q'_\Psi(\mathbf{Z}|\mathcal{G})}[\log \frac{p_\theta(\mathbf{Z}, \mathcal{G})}{q'_\Psi(\mathbf{Z}|\mathcal{G})}] \geq \mathbb{E}_{q'_\Psi(\mathbf{Z}|\mathcal{G})}\left[\log \frac{p_\theta(\mathcal{G}, \mathbf{Z})}{q_\Psi(\mathbf{Z}|\tilde{\mathcal{G}})}\right]$$

$$= \mathbb{E}_{q'_\Psi(\mathbf{Z}|\mathcal{G})}[\log p_\theta(\mathcal{G}|\mathbf{Z})] - \mathbb{E}_{q(\tilde{\mathcal{G}}|\mathcal{G})}[\mathrm{KL}(q_\Psi(\mathbf{Z}|\tilde{\mathcal{G}}))||p(\mathbf{Z})]. \tag{10}$$

The detailed derivation is in the Appendix E. Compared to VGAE, the denoising variational inference models the posterior probability $p(\mathbf{Z}|\mathcal{G})$ using a Gaussian Mixture Model, whereas VGAE models $p(\mathbf{Z}|\mathcal{G})$ using a Gaussian distribution. Additionally, during the optimization process, there is a constraint that forces $q_\Psi(\mathbf{Z}|\tilde{\mathcal{G}})$ to approximate the standard Gaussian distribution $p(\mathbf{Z})$. Consequently, our method can significantly improve the model's robustness and produce high-quality graph data. We further employ the Monte Carlo sampling method to approximate the objective, which can be effectively optimized using gradient descent as follows:

$$\mathcal{L}_d \approx \frac{1}{K} \sum_{k=1}^{K} \sum_{r\in\mathcal{R}} \log \frac{p_\theta(\mathcal{G}_r, \mathbf{Z})}{q_\Psi(\mathbf{Z}|\tilde{\mathcal{A}}_r)}, \tag{11}$$

where $K$ is the number of samples sampled during the simulation. After denoising variational inference, we input the sampled robust representations $\mathbf{Z}$ into the Heterogeneous Explanation Generator, where the complex semantics on the heterogeneous graph are learned. Before delving into that, we introduce the Graph Information Bottleneck.

## 3.3 Graph Information Bottleneck

As mentioned in the introduction, noise exacerbates the irregularity of graph structures and alters node importance. Therefore, previous methods imposing structural regularity constraints on explanatory subgraphs are infeasible under noise influence. We exploit the Graph Information Bottleneck (GIB) to enable the explainer network to adaptively handle structural irregularities. The objective of GIB is to obtain the optimal explanatory subgraph $\mathcal{G}_s$. From an information-theoretic perspective, GIB limits the amount of information carried by the explanatory subgraph $\mathcal{G}_s$, rather than imposing simple structural constraints. Simultaneously, nodes may require scattered edges across the graph to jointly explain their predictive function, rather than constraining connectedness. Consequently, GIB adaptively explores the entire graph without imposing any potentially biased restrictions. GIB can be formulated as:

$$\min_{\mathcal{G}_s \subset G} -\mathrm{I}(\hat{y}; \mathcal{G}_s) + \beta\,\mathrm{I}(\mathcal{G}; \mathcal{G}_s), \tag{12}$$

where $\mathrm{I}(\cdot; \cdot)$ denotes mutual information, and $\beta$ controls the trade-off between the two terms. Since the information in $\mathcal{G}_s$ can be continually optimized, the explain task can be characterized as an optimization task guided by GIB.

The GIB principle aims to obtain the minimum sufficient information about the graph $\mathcal{G}$. The first term maximizes the mutual information between the label and the explanatory subgraph, ensuring $\mathcal{G}_s$ contains as much information about the label as possible. The second term minimizes the mutual information between the input graph and the explanatory subgraph, ensuring $\mathcal{G}_s$ contains the minimum information about the input graph. Next, we introduce the Heterogeneous Explanation Generator, describing how each term is optimized during training under the GIB principle.

## 3.4 Heterogeneous Explanation Generator

We begin by modeling the explanatory subgraph as a Gilbert random graph [28], where edges are conditionally independent. Following the literature [16], we define an adjacency matrix-like edge matrix $E_s$, where each element $e_{ij}$ is a binary variable indicating whether the edge is included in the subgraph. When there is an edge $(i, j)$ from $v_i$ to $v_j$, $e_{ij} = 1$, otherwise $e_{ij} = 0$.

To capture the rich semantics of heterogeneous graphs, pairwise node interactions alone are insufficient. Thus, we incorporate heterogeneous semantics learning into the explanatory subgraph generation process. We incorporate heterogeneous edge type information into the attention computation by extending the standard graph attention mechanism. Specifically, we assign an edge type embedding $\mathbf{r}_{\varphi(e)}$ for each edge type $\varphi(e)$, and simultaneously utilize the edge type embeddings and node embeddings to compute the attention coefficient $\alpha_{ij}$:

$$\alpha_{ij} = \frac{\exp\left(\text{ReLU}\left(\boldsymbol{a}^T[\boldsymbol{W}\boldsymbol{z}_i\|\boldsymbol{W}\boldsymbol{z}_j\|\boldsymbol{W}_r\boldsymbol{r}_{\varphi(e_{ij})}]\right)\right)}{\sum_{k\in\mathcal{N}_i}\exp\left(\text{ReLU}\left(\boldsymbol{a}^T[\boldsymbol{W}\boldsymbol{z}_i\|\boldsymbol{W}\boldsymbol{z}_k\|\boldsymbol{W}_r\boldsymbol{r}_{\varphi(e_{ik})}]\right)\right)}, \tag{13}$$

where $\boldsymbol{W}_r$ is a learnable weight matrix for type embeddings. Edge type embedding is a one-hot encoding derived from each edge type. This attention coefficient $\alpha_{ij}$ integrates both heterogeneous node and edge type semantics, offering a more comprehensive representation.

Next, we define the heterogeneous random graph variable. The probability of the heterogeneous explanatory subgraph can be factorized as:

$$p(\mathcal{G}) = \prod_{(i,j)\in E_s} p(e_{ij}|\varphi(e_{ij})), \tag{14}$$

where $e_{ij} \sim \text{Bern}(\pi_{ij})$ and $\pi_{ij}$ is the edge existence probability inferred via $\alpha_{ij}$. To enable backpropagation through discrete edge selections, we adopt the reparameterization trick using a hard-concrete relaxation:

$$e_{ij} = \text{Sigmoid}\left(\frac{\log\epsilon - \log(1-\epsilon) + \alpha_{ij}(\varphi(e_{ij}))}{\tau}\right), \\ \epsilon \sim \text{Uniform}(0,1), \tag{15}$$

where $\tau$ is a temperature coefficient to smooth the optimization, and $\alpha_{ij}$ from Eq. 13 adds heterogeneous information into the explanatory subgraph. When $\alpha_{ij} = \log\frac{\pi_{ij}}{1-\pi_{ij}}$, we have $\lim_{\tau\to 0} p(e_{ij} = 1) = \frac{\exp(\alpha_{ij})}{1+\exp(\alpha_{ij})}$, so we can obtain the explanatory subgraph $\mathcal{G}_s$ since $p(e_{ij} = 1) = \pi_{ij}$.

This results in a continuous probability matrix $\mathbf{M_p} \in \mathbb{R}^{N\times N}$, where each entry $[\mathbf{M_p}]_{ij} = \pi_{ij}$ denotes the likelihood of including edge $(i,j)$. We then construct the soft explanatory subgraph:

$$\mathcal{G}_s = (\mathbf{A_s} = \mathbf{M_p} \odot \mathbf{A}, \mathbf{X}, \mathcal{A}_s, \mathcal{R}_s). \tag{16}$$

To optimize the explainer, we adopt the Graph Information Bottleneck (GIB) principle, balancing predictive fidelity and information compression. The GIB objective 12 is upper-bounded as:

$$-\text{I}(\hat{y};\mathcal{G}_s) + \beta\,\text{I}(\mathcal{G};\mathcal{G}_s) \leq -\mathbb{E}_{p(\mathcal{G}_s,\hat{y})}\big[\log p_f(\hat{y}|\mathcal{G}_s)\big] + \text{H}(\hat{y}) + \beta\mathbb{E}_{p(\mathcal{G})}\big[\text{KL}(p_\alpha(\mathcal{G}_s|\mathcal{G})\|q(\mathcal{G}_s))\big], \tag{17}$$

where $f$ is the GNN model and $\alpha$ is the explain model, see Appendix E for detailed derivation. Since $\text{H}(\hat{y})$ is constant, the objective function can be expressed as follows:

$$\mathcal{L}_{GIB} = -\mathbb{E}_{p(\mathcal{G}_s,\hat{y})}\big[\log p_f(\hat{y}|\mathcal{G}_s)\big] + \beta\mathbb{E}_{p(\mathcal{G})}\big[\text{KL}(p_\alpha(\mathcal{G}_s|\mathcal{G})\|q(\mathcal{G}_s))\big]. \tag{18}$$

The total loss combines the GIB objective with the denoising loss from the variational graph encoder:

$$\mathcal{L} = \mathcal{L}_d + \mathcal{L}_{GIB}. \tag{19}$$

### 3.5 Complexity Analysis.

The cost of each iteration comprises two parts: (1) the variational inference process and (2) the heterogeneous explanation generation. The time complexity of the first step is $O(N^2 + E)$, and the space complexity is $O(N)$, as this step requires storing the robust node representations. The time complexity of the second step is $O(E)$, and the space complexity is $O(E)$. Therefore, the overall time complexity of RoHeX is $O(N^2 + E)$, and the space complexity is $O(N + E)$.

## 4 Experiment

In this section, we evaluate the performance of the proposed RoHeX and state-of-the-art baselines on the node classification task. We then analyze the contributions of different components of RoHeX and demonstrate why RoHeX is robust to noise and capable of generating explanations that incorporate heterogeneous information.

Table 1: The comparison of RoHeX and baselines under different ratios of random structural noise. We use **bold** font to mark the best score. The second best score is marked with underline.

| Dataset | Metric | Noise | PGExplainer | GNNExplainer | PGM-Explainer | V-InfoR | AMExplainer | Hete-PGE | xPath | RoHeX |
|---|---|---|---|---|---|---|---|---|---|---|
| DBLP | MAE | 10% | 1.2158±0.0062 | 0.8530±0.0009 | 1.0704±0.0007 | 1.1930±0.0030 | 1.4862±0.0026 | 0.8719±0.0008 | 0.8162±0.0032 | **0.8359±0.0029** |
| | | 20% | 1.2179±0.0089 | 0.9080±0.0007 | 1.2046±0.0006 | 1.1960±0.0025 | 1.5697±0.0008 | 0.8896±0.0005 | 0.9651±0.0042 | **0.8743±0.0018** |
| | | 30% | 1.2449±0.0059 | 1.2613±0.0011 | 1.3313±0.0002 | 1.2312±0.0026 | 1.7133±0.0006 | 1.1913±0.0006 | 1.1405±0.1032 | **0.8827±0.0034** |
| | | 40% | 1.2451±0.0068 | 1.3389±0.0008 | 1.3401±0.0005 | 1.2530±0.0010 | 1.9345±0.0005 | 0.9268±0.0006 | 1.2852±0.0002 | **0.9014±0.0015** |
| | RMSE | 10% | 1.6775±0.0054 | 1.2968±0.0006 | 1.3855±0.0005 | 1.6481±0.0027 | 1.9219±0.0048 | 1.2814±0.0004 | 1.3175±0.0020 | **1.2416±0.0017** |
| | | 20% | 1.6815±0.0666 | 1.3072±0.0004 | 1.5280±0.0003 | 1.6511±0.0020 | 2.0419±0.0006 | 1.2859±0.0002 | 1.4341±0.0025 | **1.2750±0.0013** |
| | | 30% | 1.6999±0.0024 | 1.8470±0.0007 | 1.6497±0.0001 | 1.6781±0.0020 | 2.1842±0.0003 | 1.6532±0.0003 | 1.5752±0.1198 | **1.2792±0.0022** |
| | | 40% | 1.7060±0.0042 | 1.9043±0.0004 | 1.6681±0.0001 | 1.6885±0.0008 | 2.3616±0.0002 | 1.3100±0.0004 | 1.6819±0.0001 | **1.2889±0.0008** |
| ACM | MAE | 10% | 0.7624±0.0080 | 0.3449±0.0003 | 0.2155±0.0009 | 0.7639±0.0001 | 0.3895±0.0005 | 0.8091±0.0003 | 0.3900±0.0001 | **0.2129±0.0009** |
| | | 20% | 0.7751±0.0162 | 0.3951±0.0001 | 0.3732±0.0003 | 0.7913±0.0004 | 0.6746±0.0224 | 0.8183±0.0005 | 0.3985±0.0003 | **0.2483±0.0010** |
| | | 30% | 0.7867±0.0152 | 0.5087±0.0003 | 0.5932±0.0006 | 0.8064±0.0003 | 0.7077±0.0221 | 0.8220±0.0002 | 0.4164±0.0012 | **0.3140±0.0019** |
| | | 40% | 0.7913±0.0181 | 0.6496±0.0002 | 0.7932±0.0007 | 0.8154±0.0005 | 0.7181±0.0164 | 0.8310±0.0001 | 0.4292±0.0006 | **0.3163±0.0015** |
| | RMSE | 10% | 1.0258±0.0037 | 0.6831±0.0005 | **0.5121±0.0004** | 1.0145±0.0002 | 0.6241±0.0003 | 1.0740±0.0001 | 0.7750±0.0002 | 0.5662±0.0012 |
| | | 20% | 1.0307±0.0069 | 0.7791±0.0002 | 0.6893±0.0002 | 1.0366±0.0004 | 0.8213±0.0649 | 1.0778±0.0004 | 0.7458±0.0003 | **0.6177±0.0008** |
| | | 30% | 1.0423±0.0127 | 0.9506±0.0001 | 0.8793±0.0004 | 1.0705±0.0007 | 0.8412±0.0696 | 1.0731±0.0001 | 0.7768±0.0009 | **0.6669±0.0013** |
| | | 40% | 1.0431±0.0124 | 1.1306±0.0002 | 1.0766±0.0004 | 1.0786±0.0003 | 0.8474±0.0713 | 1.0856±0.0001 | 0.8396±0.0007 | **0.6909±0.0008** |
| Freebase | MAE | 10% | 0.7189±0.0096 | 0.9012±0.0002 | 0.9190±0.0003 | 0.5957±0.0322 | 0.9312±0.0004 | 0.7760±0.0007 | 0.9006±0.0127 | **0.3885±0.0010** |
| | | 20% | 0.7237±0.0078 | 0.9108±0.0003 | 0.9401±0.0001 | 0.6822±0.0527 | 0.9709±0.1209 | 0.7812±0.0005 | 0.9247±0.0092 | **0.4441±0.0012** |
| | | 30% | 0.7285±0.0041 | 0.9126±0.0003 | 0.9530±0.0004 | 0.7249±0.0329 | 1.0089±0.1651 | 0.7908±0.0003 | 0.9263±0.0043 | **0.4694±0.0021** |
| | | 40% | 0.7370±0.0019 | 0.9378±0.0007 | 0.9587±0.0009 | 0.7894±0.0111 | 1.0531±0.0005 | 0.8030±0.0001 | 0.9301±0.0061 | **0.4880±0.0014** |
| | RMSE | 10% | 1.0616±0.0071 | 1.2886±0.0001 | 1.2432±0.0001 | 1.0375±0.0233 | 1.2589±0.0004 | 1.1064±0.0003 | 1.2466±0.0056 | **0.8251±0.0018** |
| | | 20% | 1.0635±0.0051 | 1.2983±0.0002 | 1.2549±0.0001 | 1.1172±0.0566 | 1.2987±0.1848 | 1.1117±0.0004 | 1.2435±0.0108 | **0.8854±0.0010** |
| | | 30% | 1.0689±0.0039 | 1.2995±0.0002 | 1.2747±0.0002 | 1.1487±0.0277 | 1.3391±0.2381 | 1.1200±0.0002 | 1.2721±0.0034 | **0.9035±0.0012** |
| | | 40% | 1.0803±0.0018 | 1.3217±0.0005 | 1.2838±0.0004 | 1.1929±0.0054 | 1.3830±0.0004 | 1.1315±0.0001 | 1.2643±0.0072 | **0.9217±0.0007** |

## 4.1 Experiment Settings

**Datasets and Baselines.** We evaluate the effectiveness of our RoHeX on three real-world datasets, including two academic citation datasets (DBLP and ACM) and a knowledge graph dataset (Freebase). Since there are no existing robust heterogeneous explainer, we select three types of baselines: the surrogate method PGM-Explainer, the perturbation-based methods GNNExplainer, PGExplainer and AMExplainer, and the V-infor method studying robustness on homogeneous graphs. We used the heterogeneous graph path explainer, xPath, and extended our Heterogeneous Explanation Generator to PGExplainer, referred to as Hete-PGE, for comparison.

**Evaluation.** The evaluation of explainer performance is based on the generated explanatory subgraphs, assessing their contribution to the original prediction. We adopt two metrics: fidelity and sparsity. Fidelity measures the decrease in prediction confidence after removing the explanation from the input graph, while sparsity measures the ratio of remaining edges in the explanatory subgraph $\mathcal{G}_s$ relative to the input graph. We use the Mean Absolute Error (MAE, $\frac{1}{N}\sum_{i=1}^{N}\left|\mathbb{I}(\hat{y}_i = y_i) - \mathbb{I}(\hat{y}_i^{\mathcal{G}_s} = y_i)\right|$),

and Root Mean Squared Error (RMSE, $\sqrt{\frac{1}{N}\sum_{i=1}^{N}\left(\mathbb{I}(\hat{y}_i = y_i) - \mathbb{I}(\hat{y}_i^{\mathcal{G}_s} = y_i)\right)^2}$) as proxy measures for fidelity, and compare the performance of different baselines across varying sparsity levels, where $N$ is the number of nodes or graphs, $\hat{y}_i$ is the original prediction result, and $\hat{y}_i^{\mathcal{G}_s}$ is the prediction result obtained by the explanatory subgraph.

**Implementation Details.** We conduct experiments under different proportions of random noise scenarios. Noise is added to both the training set and the test set to restore the real scene. To ensure randomness, the deletion rate and false addition rate are set to be equal. For the baselines, we select the best-performing parameters for heterogeneous datasets based on the original settings. We chose the most basic HGNN architecture, which only includes GCN [29] and relational learning modules, as the base model for fair comparison. For our RoHeX, we use Adam as the optimizer with a learning rate of 1e-4. We set the hidden dimension for variational inference to 64, the output dimension to 32, and the edge weight output dimension to 32. Each experiment is repeated 5 times, and we report the mean and variance as the results. Descriptions of the variance, datasets, baselines, base HGNN model architecture, and parameter settings are provided in the Appendix F.

## 4.2 Overall Performance under Structural Perturbations

Table 1 shows the experimental results on the heterogeneous graphs with different budgets of structural perturbations. We find that RoHeX outperforms other baselines in most experimental results, achieving the best performance on the DBLP and Freebase datasets. Taking 30% noise ratio as an example, RoHeX shows 25.9% lower MAE and 22.4% lower RMSE than the second-best method on the DBLP dataset, 38.2% lower MAE and 24.1% lower RMSE on the ACM dataset, and 35.2% lower MAE and 15.4% lower RMSE on the Freebase dataset. xPath performs excellently in multiple

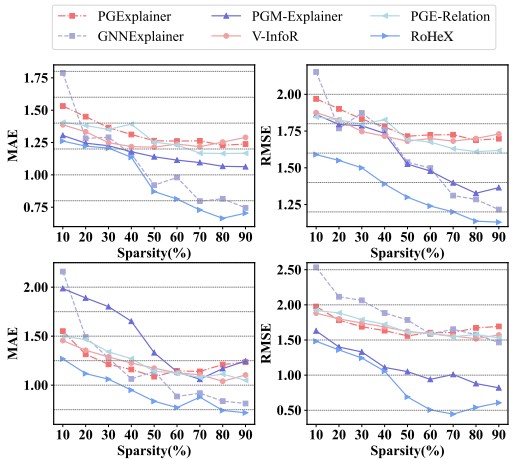

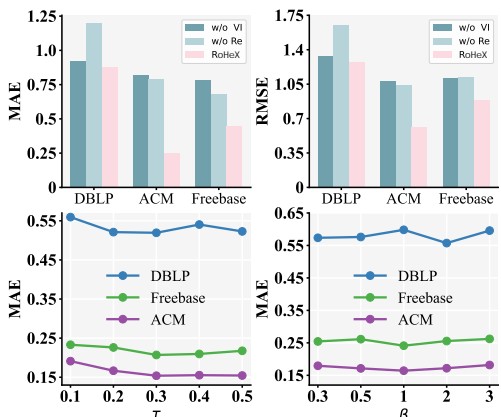

Figure 2: Fidelity-Sparsity Curve on the DBLP dataset. The first row is the result without noise, and the second row is the result with 20% noise budget.

Figure 3: Ablation study on three datasets and the influence of hyperparameters $\tau$ and $\beta$ on RoHeX.

scenarios, indicating that designing corresponding modules for heterogeneous graphs is essential. We can observe that Hete-PGE achieves second best performance multiple times on the DBLP dataset and outperforms many baselines on other datasets, demonstrating the effectiveness of our proposed heterogeneous explanation generator in considering rich semantics on heterogeneous graphs. Due to the similar edge type distribution in the DBLP dataset, the dataset exhibits higher heterogeneity, which enhances the module's ability to capture heterogeneous information. Simultaneously, as a plug-and-play module, it can be conveniently extended to other parameterized explanation methods for generating explanations on heterogeneous graphs. On the medium and small-scale datasets DBLP and ACM, explanation methods based on raw features (e.g., GNNExplainer) are more susceptible to noise, potentially because raw features are more easily affected in smaller graphs. Since RoHeX generates robust graph representations, it can better mitigate the influence of noise, which is also why the latent representation-based explainer V-InfoR performs well in multiple scenarios. Under the guidance of GIB, our method can adaptively select important edges while excluding redundant and noisy edges, thereby generating the best explanations for the prediction model.

### 4.3 Fidelity-Sparsity Analysis

Next, we further investigate RoHeX's performance at different sparsity levels. We provide the Fidelity-Sparsity curve on the DBLP dataset as shown in Figure 2. RoHeX consistently outperforms other baselines across all sparsity levels, indicating that our method can generate the best explanations. As the sparsity increases from 0 to 1, the overall trend of all curves is downward, i.e., decreasing error. When the sparsity is extremely low, e.g., 10%, our method significantly outperforms other baselines, suggesting that RoHeX can identify the truly critical subgraphs. We further find that although the overall performance improves as the sparsity level increases, there are still some cases where the performance drops with increasing sparsity, such as PGExplainer. We conjecture that this may be because in the subgraph generation process, when the sparsity increases to a point where all edges with high importance scores have been selected, forcing higher sparsity will begin to select unimportant edges, which can be viewed as noisy edges, leading to degraded performance. As the sparsity continues to increase, this adverse effect is offset. Since AMExplainer and xPath contain specific Sparsity settings to generate explanations, they were not included in the experiment.

### 4.4 Model Analysis

**Perturbation Performance** We analyze the perturbation performance of controllable structural perturbation for heterogeneous graphs under different noise scenarios, as shown in Table 2. Compared to the original results, our structural perturbation method significantly impacts the decision-making process of HGNN on the whole graph (Noisy), 1-hop subgraph (1-hop), and 2-hop subgraph (2-hop).

Table 2: Prediction performance of HGNN in different noise scenarios.

| Dataset | Method | 10% | | 20% | | 30% | | 40% | |
|---------|--------|---------|---------|---------|---------|---------|---------|---------|---------|
| | | Micro-F1 | Macro-F1 | Micro-F1 | Macro-F1 | Micro-F1 | Macro-F1 | Micro-F1 | Macro-F1 |
| DBLP | Original | 92.64 | 92.16 | 92.64 | 92.16 | 92.64 | 92.16 | 92.64 | 92.16 |
| | Noisy | 81.58 | 80.34 | 73.41 | 70.26 | 67.18 | 62.02 | 62.99 | 55.91 |
| | 1-hop | 57.28 | 55.92 | 54.22 | 52.40 | 52.18 | 49.98 | 50.56 | 48.40 |
| | 2-hop | 57.64 | 55.65 | 42.07 | 38.55 | 41.76 | 38.06 | 39.01 | 34.01 |
| ACM | Original | 92.32 | 92.40 | 92.32 | 92.40 | 92.32 | 92.40 | 92.32 | 92.40 |
| | Noisy | 33.52 | 19.28 | 33.28 | 18.82 | 31.96 | 16.14 | 31.96 | 16.14 |
| | 1-hop | 77.10 | 77.31 | 76.39 | 76.51 | 75.40 | 75.57 | 67.94 | 67.81 |
| | 2-hop | 52.19 | 51.73 | 49.12 | 50.65 | 42.38 | 36.42 | 38.73 | 30.07 |
| Freebase | Original | 68.99 | 63.57 | 68.99 | 63.57 | 68.99 | 63.57 | 68.99 | 63.57 |
| | Noisy | 67.89 | 61.90 | 66.05 | 59.29 | 62.94 | 53.74 | 57.50 | 46.02 |
| | 1-hop | 58.52 | 53.23 | 57.62 | 51.94 | 57.50 | 51.21 | 57.01 | 51.14 |
| | 2-hop | 44.66 | 31.12 | 42.90 | 27.24 | 41.10 | 24.15 | 39.46 | 21.62 |

An interesting finding is that 30% noise is enough to cause the performance of HGNN on the ACM dataset to drop to its lowest.

**Ablation Study**    We investigate the contributions of different components in RoHeX. Specifically, we study (a) the effectiveness of the denoising variational inference module, and (b) the effectiveness of the relation-based importance module. We use 'w/o VI' to denote the model without the denoising variational inference module, and 'w/o Re' to denote the model without the relation-based importance module. For the latter case, we replace it with the common concatenation operation, i.e., $\alpha_{ij} = \mathrm{MLP}[(\mathbf{z}_i, \mathbf{z}_j)]$. The experiments are conducted under 20% noise budget, and the first row of Figure 3 shows the results after ablation. We find that without the denoising variational inference module, the model relies on the original features and graph structure for prediction, failing to mitigate the influence of noise, leading to performance degradation. When the model loses the ability to learn heterogeneous relationships, the process of generating explanation subgraphs struggles to recognize the complex semantics in heterogeneous graphs. All edges are treated as the same type, and the model explains solely based on node interactions. This demonstrates the necessity of our proposed relation importance module.

**Hyperparameter Analysis**    We further analyze the impact of two parameters $\tau$ and $\beta$ on model performance. $\tau$ controls the approximation degree of $e_{ij}$ distribution to the Bernoulli distribution, ranging within $[0.1, 0.5]$. $\beta$ balances the information recovery strength and information filtering strength in the optimization objective, and we select values from $\{0.3, 0.5, 1, 2, 3\}$. The second row of Figure 3 shows the effects of these hyperparameters on RoHeX across three datasets. We can observe that the best results of $\tau$ all appear around 0.3. That is, when $\tau = 0.3$, the continuity and approximation degree in Eq. 15 reach the best trade-off. Secondly, RoHeX is not very sensitive to $\beta$ that controls the constraint strength in Eq. 18, validating that our used GIB constraint can adapt to different data scenarios and achieve superior performance.

# 5   Conclusion

In this work, we focus on the problem of explaining heterogeneous graph neural network under noise. We are the first to study this problem, theoretically proving that heterogeneous graph neural network have an amplifying effect on noise, and propose RoHeX to mitigate the influence of noise and obtain explanatory subgraphs based on heterogeneous relations. Specifically, RoHeX employs denoising variational inference to obtain robust graph representations and parameterizes the explanatory subgraph generation process with heterogeneous semantics. It integrates type information to capture the complexity of diverse node and edge types. Extensive experiments on real-world datasets demonstrate RoHeX's superiority over other state-of-the-art baselines. For future work, we plan to further extend RoHeX to dynamic graphs by incorporating dynamic information into the explanation generation process, further broadening RoHeX's applicability.

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

## A  Related Work

**GNN Explainability.**   Recently, various approaches have been proposed to explain the predictions of GNN, these approaches can be categorized into post-hoc and built-in method. Common post-hoc methods include perturbation-based [6, 30] and surrogate model-based [7, 8] approaches. MixupExplainer [31] extends the existing GIB framework by introducing label-independent subgraphs during the sampling of explanation subgraphs, thereby obtaining explanations while mitigating the distribution shift phenomenon. GNNExplainer [32] learns masks for features and edges by optimizing the masks to obtain the optimal explanation. PGExplainer [16] employs a parametric neural network approach to learn the importance of each edge and ultimately selects edges with high importance scores to construct the explanatory subgraph. PGM-Explainer [33] adopts a Bayesian network formulation, naturally expressing the dependencies between nodes in the form of conditional probabilities. Common built-in methods include prototype learning-based [9, 10] and graph generation-based [11] approaches. PGIB [9] integrates prototypes into the Graph Information Bottleneck framework, allowing it to learn prototypes based on key subgraphs in the input graph, thereby providing a more accurate explanation of the prediction process. GOAt [34] generates explanatory subgraphs by decomposing the model's output into a series of scalar products involving node and edge features, and calculating the contribution of each feature to these scalar products, thereby highlighting the edges that are important for the prediction outcome. xPath [35] provides fine-grained explanations by identifying cause nodes and their influence paths through a novel graph rewiring algorithm, thereby offering detailed insights into how specific nodes affect model predictions. AMExplainer [36] leverages adversarial networks to optimize for both sparsity and prediction accuracy in explanations, significantly enhancing the clarity and efficiency of model interpretability.

**Heterogeneous Graph Neural Networks**   Heterogeneous Graph Neural Networks can be categorized into meta-path-based methods and neighborhood aggregation-based methods. Meta-path-based methods typically decompose heterogeneous graphs into multiple homogeneous subgraphs using predefined meta-paths, thereby capturing specific types of heterogeneous semantics. Message passing is then performed within each subgraph, and the messages are subsequently aggregated. Common methods in this category include HAN [37], MAGNN [38], and SeHGNN [39]. On the other hand, neighborhood aggregation-based methods usually aggregate information directly from neighbors and apply specific aggregation strategies based on node types. Examples of methods in this category include RGCN [40], NARS [41], and Simple-HGN [24].

## B  Proof of Theorem 3.1

Let node $v_i$'s original degree under edge type $r$ be denoted as $d_i^r$. The deletion rate for edge type $r$ is $\eta_r^+$, and the false addition rate is $\eta_r^-$. The average degree for edge type $r$ is defined as:

$$\bar{d}^r = \frac{1}{|\mathcal{V}_r|} \sum_{i \in \mathcal{V}_r} d_i^r \tag{20}$$

where $\mathcal{V}_r$ represents the set of nodes involved in edge type $r$. For node $v_i$, the expected number of edges deleted under edge type $r$ is:

$$\mathbb{E}[\Delta d_{i,+}^r] = \sum_{j \in N_r(i)} \eta_r^+ \cdot \frac{d_i^r + d_j^r}{2\bar{d}^r} \tag{21}$$

where $\mathcal{N}_r(i)$ represents the set of neighbors of node $v_i$ under edge type $r$. Assuming the degree distribution of neighbors is uniform, it can be approximated as:

$$\mathbb{E}[\Delta d_{i,+}^r] \approx d_i^r \cdot \eta_r^+ \cdot \frac{d_i^r + \bar{d}^r}{2\bar{d}^r} \tag{22}$$

Further simplification gives:

$$\mathbb{E}[\Delta d_{i,+}^r] \approx \eta_r^+ \cdot \frac{d_i^{r\,2}}{2\bar{d}^r} \tag{23}$$

For node $v_i$, the expected number of false added edges under edge type $r$ is:

$$\mathbb{E}[\Delta d_{i,-}^r] = \sum_{j \notin \mathcal{N}_r(i)} \eta_r^- \cdot \frac{1}{d_i^r + d_j^r} \tag{24}$$

Assuming the degree distribution of non-neighbor nodes is uniform, it can be approximated as:

$$\mathbb{E}[\Delta d_{i,-}^r] \approx (|\mathcal{V}_r| - d_i^r) \cdot \eta_r^- \cdot \frac{1}{d_i^r + \bar{d}^r} \tag{25}$$

For node $v_i$, the total degree change across all edge types is:

$$\mathbb{E}[\Delta d_i] = \sum_{r \in R} (\mathbb{E}[\Delta d_{i,-}^r] - \mathbb{E}[\Delta d_{i,+}^r]) \tag{26}$$

Substituting the above results, we get:

$$\mathbb{E}[\Delta d_i] = \sum_{r \in \mathcal{R}} \left( \eta_r^- \cdot \frac{(|\mathcal{V}_r| - d_i^r)}{d_i^r + \bar{d}^r} - \eta_r^+ \cdot \frac{d_i^{r\,2}}{2\bar{d}^r} \right) \tag{27}$$

The expected degree of node $v_i$ after perturbation is:

$$\mathbb{E}[d_i'] = d_i + \mathbb{E}[\Delta d_i] \tag{28}$$

Substituting the expression for $\mathbb{E}[\Delta d_i]$, we get:

$$\mathbb{E}[d_i'] = d_i + \sum_{r \in \mathcal{R}} \left( \eta_r^- \cdot \frac{(|\mathcal{V}_r| - d_i^r)}{d_i^r + \bar{d}^r} - \eta_r^+ \cdot \frac{d_i^{r\,2}}{2\bar{d}^r} \right) \tag{29}$$

To maintain the degree distribution characteristics of the graph, we introduce a degree-balancing constraint, which states that the expected degree after perturbation should be as close as possible to the original degree. Specifically, we require:

$$|\mathbb{E}[d_i'] - d_i| \leq \epsilon \tag{30}$$

where $\epsilon$ is a small constant representing the allowed degree deviation. By adjusting the deletion rate $\eta_r^+$ and false addition rate $\eta_r^-$, we can satisfy this constraint.

The above derivation shows that the degree-balancing constraint can effectively control the impact of noise injection on the graph structure. For high-degree nodes, the deletion rate $\eta_r^+$ has a larger weight $\frac{d_{ir}}{2\bar{d}_r}$, thereby reducing the number of edges to be deleted. For low-degree nodes, the false addition rate $\eta_r^-$ has a larger weight $\frac{1}{d_{ir}+\bar{d}_r}$, thereby increasing the number of edges to be added. This design ensures that the perturbed graph structure maintains degree distribution characteristics similar to the original graph, thereby improving the reasonableness of noise injection.

Table 3: Statistics of the graphs before and after perturbation.

| Dataset | Budget | $\triangle \bar{d}$ | JS Divergence |
|---------|--------|------|---------------|
| DBLP | 10% | 0.0831 | 0.0875 |
| | 20% | 0.8337 | 0.0905 |
| | 30% | 1.7506 | 0.1114 |
| | 40% | 2.6675 | 0.1407 |
| ACM | 10% | 4.0125 | 0.2156 |
| | 20% | 9.0189 | 0.2126 |
| | 30% | 14.0254 | 0.2131 |
| | 40% | 19.0318 | 0.2188 |
| Freebase | 10% | 0.6556 | 0.2029 |
| | 20% | 0.3112 | 0.1516 |
| | 30% | 0.0332 | 0.1167 |
| | 40% | 0.3776 | 0.0897 |

## C  Effectiveness of Controllable Structural Perturbation and Theorem 3.1

We propose controllable structural perturbation on heterogeneous graphs to simulate real-world noise. Our focus is on statistical perturbation effects rather than individual node changes. The

Table 4: The performance of RoHeX under different distribution perturbations.

| Dataset | Budget | 10% | | 20% | | 30% | | 40% | |
|---------|--------|------|------|------|------|------|------|------|------|
| | | MAE | RMSE | MAE | RMSE | MAE | RMSE | MAE | RMSE |
| DBLP | Uniform | 0.7940 | 1.3462 | 0.8877 | 1.3708 | 0.9391 | 1.4185 | 0.9486 | 1.3891 |
| | Real | 0.8359 | 1.2416 | 0.8743 | 1.2750 | 0.8827 | 1.2792 | 0.9014 | 1.2889 |
| | Difference | 5.01% | -8.42% | -1.53% | -7.51% | -6.39% | -10.89% | -5.24% | -7.77% |
| ACM | Uniform | 0.2225 | 0.5648 | 0.2544 | 0.5986 | 0.3362 | 0.7516 | 0.3743 | 0.7333 |
| | Real | 0.2129 | 0.5662 | 0.2483 | 0.6177 | 0.3140 | 0.6669 | 0.3163 | 0.6909 |
| | Difference | -4.51% | 0.25% | -2.46% | 3.09% | -7.07% | -12.70% | -18.34% | -6.14% |
| Freebase | Uniform | 0.4331 | 0.7906 | 0.4564 | 0.9125 | 0.5186 | 0.9758 | 0.5247 | 0.9818 |
| | Real | 0.3885 | 0.8251 | 0.4441 | 0.8854 | 0.4694 | 0.9035 | 0.4880 | 0.9217 |
| | Difference | -11.47% | 4.18% | -2.78% | -3.06% | -10.48% | -8.01% | -7.51% | -6.53% |

uniform assumption was introduced to simplify the derivation, a practice widely adopted in prior works [42, 43]. Assuming uniform distribution effectively captures perturbation impacts. To further validate this, we analyzed the degree distributions of the DBLP, ACM, and Freebase datasets before and after perturbation, as shown in Table 3. The results indicate that our method achieves significant noise effect while only slightly altering the degree distribution (with JS divergence $< 0.22$), as demonstrated in Table 2 of the main text.

Additionally, we conducted experiments comparing the performance of the explainer under the uniform assumption and the actual degree distribution. The results, presented in Table 4, confirm that the uniform assumption has minimal impact on model performance compared to using the real distribution.

## D  Proof of Theorem 3.2

In Graph Neural Network, a node representation is typically updated by aggregating information from its neighboring nodes. This process can be described as a message passing mechanism, where each node receives messages from neighboring nodes and updates its representation based on these messages. To avoid cases where the influence is overly amplified during the aggregation process, the messages from neighboring nodes are typically normalized. A common normalization approach is to multiply each neighbor message by the inverse of its degree. Assuming that each node influence on neighbors is equal, a higher-degree node will distribute its influence evenly among all neighbors. Therefore, the influence received by each neighbor should be proportional to the inverse of the node degree. In contrast, in random walk models, the transition probability between nodes is inversely proportional to the node degree. That is, the probability of a node reaching a particular neighbor is the inverse of its degree.

Given a heterogeneous graph $\mathcal{G}$, let $v_i$ be a node with degree $d_{v_i}$. A noisy edge $e_{ij}$ is added to the graph, where $v_j$ is a new neighbor with degree $d_{v_j}$ and $k$ specific-type neighbors that match a given meta-path $\phi$.

For meta-path-based methods:

(a) Before adding the noisy edge, the influence of $v_i$ is assumed to be a combination of the influences from its $d_{v_i}$ existing neighbors $v_1, v_2, ..., v_{d_{p_i}}$ in $\mathcal{G}$. The influence of each neighbor $v_n$ on $v_i$ can be represented as:

$$I_{\text{ori1}} = \sum_{n=1}^{d_{v_i}} \frac{1}{d_{v_i}} \tag{31}$$

(b) After adding the noisy edge $(v_i, v_j)$, $v_i$ is directly connected to $v_j$, and the influence of $v_j$ will propagate to its $k$ neighbors. The influence on each neighbor of $v_j$ changes in the following manner $v_i$:

$$I_{\text{new1}} = \sum_{i=1}^{d_{v_i}} \frac{1}{d_{v_i}+1} + \frac{k}{d_{v_i}+1} = \frac{d_{v_i}}{d_{v_i}+1} + \frac{k}{d_{v_i}+1} = \frac{d_{v_i}+k}{d_{v_i}+1} \tag{32}$$

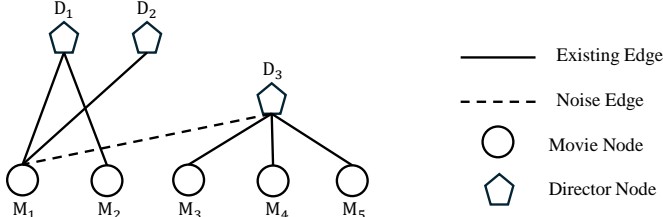

Figure 4: The illustrative example for noise against HGNNs on Movie-Director graph.

For neighborhood aggregation-based methods:

(a) Before adding the noisy edge, the influence on the direct neighbors of $v_i$ is given by:

$$I_{\text{ori2}} = \sum_{n=1}^{d_{v_i}} \frac{1}{d_{v_i}} \tag{33}$$

(b) After adding the noisy edge $(v_i, v_j)$, the neighbors of $v_i$ increase to $d_{v1} + 1$, and the influence on each of its neighbors changes to:

$$I_{\text{new2}} = \sum_{n=1}^{d_{v_i}+1} \frac{1}{d_{v_i} + 1} \tag{34}$$

The multiplicative relationship $\xi$ of the influence propagation between the meta-path-based method and the neighborhood aggregation method is:

$$\xi = \frac{\frac{I_{\text{new1}}}{I_{\text{ori1}}}}{\frac{I_{\text{new2}}}{I_{\text{ori2}}}} = \frac{\frac{d_{v_i}+k}{d_{v_i}+1}}{1} = \frac{d_{v_i} + k}{d_{v_i} + 1} \tag{35}$$

Consequently, when $k > d_{v_i}$, the multiplicative factor $\xi$ is significantly greater than 1. This indicates that in general heterogeneous graphs, meta-path-based approaches are far more susceptible to the influence of noisy edges compared to neighborhood aggregation-based approaches. This substantiates that meta-path-based methods can significantly amplify the effect of noisy edges to a greater extent than neighborhood aggregation methods.

We also provide the illustrative example for noise against HGNNs on Movie-Director graph in Figure 4. The meta-path used is M-D-M. In neighborhood aggregation-based HGNNs, the noise-introduced nodes are not directly considered as 1-hop neighbors of $M_1$. Consequently, under the influence of noise, the 2-hop neighbors $M_3, M_4, M_5$ can only affect $M_1$ through the 1-hop neighbor $D_3$. However, in meta-path-based HGNNs, all neighbors under the M-D-M meta-path are aggregated with equal weight $\frac{1}{5}$, thereby enlarging the effect of the noisy edge $\langle M_1, D_3 \rangle$ to $\frac{3}{5}$ (the total weight of the noisy neighbors $M_3, M_4, M_5$).

## E    Detailed Derivation

First, we give the detailed derivation of Eq. 7. We introduce the Kullback-Leibler (KL) divergence. The KL divergence is a measure used to quantify the difference between two probability distributions. Let us consider two continuous random variables with probability distributions $P$ and $Q$, and their corresponding probability density functions denoted as $p(x)$ and $q(x)$, respectively. If we aim to approximate $p(x)$ using $q(x)$, the KL divergence can be expressed as:

$$\text{KL}(P||Q) = \int p(x) \log \frac{p(x)}{q(x)} dx. \tag{36}$$

Because the logarithmic function is convex, the value of KL divergence is nonnegative. Then, Eq. 7 can be written as:

$$
\begin{aligned}
\mathcal{L}(\Psi, \theta; \mathcal{G}) &= \mathbb{E}_{q_\Psi(\mathbf{Z}|\mathcal{G})}[\log \frac{p_\theta(\mathbf{Z}, \mathcal{G})}{q_\Psi(\mathbf{Z}|\mathcal{G})}] \\
&= \mathbb{E}_{q_\Psi(\mathbf{Z}|\mathcal{G})}[\log p_\theta(\mathbf{Z}|\mathcal{G}) \cdot \frac{p(\mathbf{Z})}{q_\Psi(\mathbf{Z}|\mathcal{G})}] \\
&= \mathbb{E}_{q_\Psi(\mathbf{Z}|\mathcal{G})}[\log p_\theta(\mathbf{Z}|\mathcal{G})] - \mathrm{KL}(q_\Psi(\mathbf{Z}|\mathcal{G})||p(\mathbf{Z})).
\end{aligned}
\tag{37}
$$

Second, the lower bound of denoising variational inference in Eq. 10 can be derived as:

$$
\begin{aligned}
\mathcal{L}_d &= \mathbb{E}_{q'_\Psi(\mathbf{Z}|\mathcal{G})}[\log \frac{p_\theta(\mathbf{Z}, \mathcal{G})}{q'_\Psi(\mathbf{Z}|\mathcal{G})}] \geq \mathbb{E}_{q'_\Psi(\mathbf{Z}|\mathcal{G})}\left[\log \frac{p_\theta(\mathcal{G}, \mathbf{Z})}{q_\Psi(\mathbf{Z}|\tilde{\mathcal{G}})}\right] \\
&= \mathbb{E}_{q'_\Psi(\mathbf{Z}|\mathcal{G})}[\log p_\theta(\mathcal{G}|\mathbf{Z}) + \log p(\mathbf{Z}) - \log q_\Psi(\mathbf{Z}|\tilde{\mathcal{G}})] \\
&= \mathbb{E}_{q'_\Psi(\mathbf{Z}|\mathcal{G})}[\log p_\theta(\mathcal{G}|\mathbf{Z})] - \mathbb{E}_{q'_\Psi(\mathbf{Z}|\mathcal{G})}\left[\log \frac{q_\Psi(\mathbf{Z}|\tilde{\mathcal{G}})}{p(\mathbf{Z})}\right] \\
&= \mathbb{E}_{q'_\Psi(\mathbf{Z}|\mathcal{G})}[\log p_\theta(\mathcal{G}|\mathbf{Z})] - \mathbb{E}_{q(\tilde{\mathcal{G}}|\mathcal{G})}\mathbb{E}_{q_\Psi(\mathbf{Z}|\mathcal{G})}\left[\log \frac{q_\Psi(\mathbf{Z}|\tilde{\mathcal{G}})}{p(\mathbf{Z})}\right] \\
&= \mathbb{E}_{q'_\Psi(\mathbf{Z}|\mathcal{G})}[\log p_\theta(\mathcal{G}|\mathbf{Z})] - \mathbb{E}_{q(\tilde{\mathcal{G}}|\mathcal{G})}[\mathrm{KL}(q_\Psi(\mathbf{Z}|\tilde{\mathcal{G}}))||p(\mathbf{Z})].
\end{aligned}
\tag{38}
$$

Third, we derive an upper bound for GIB in Eq. 17. We decompose the mutual information:

$$
\mathrm{I}(\hat{y}; \mathcal{G}_s) = \mathbb{E}_{p(\hat{y}, \mathcal{G}_s)}\left[\log \frac{p(\hat{y}, \mathcal{G}_s)}{p(\hat{y})p(\mathcal{G}_s)}\right], \mathrm{I}(\mathcal{G}; \mathcal{G}_s) = \mathbb{E}_{p(\mathcal{G}, \mathcal{G}_s)}\left[\log \frac{p(\mathcal{G}, \mathcal{G}_s)}{p(\mathcal{G})p(\mathcal{G}_s)}\right].
\tag{39}
$$

The GIB objective can be written as:

$$
\begin{aligned}
&- \mathrm{I}(\hat{y}; \mathcal{G}_s) + \beta\, \mathrm{I}(\mathcal{G}; \mathcal{G}_s) \\
&= -\mathbb{E}_{p(\hat{y}, \mathcal{G}_s)}\left[\log \frac{p(\hat{y}, \mathcal{G}_s)}{p(\hat{y})p(\mathcal{G}_s)}\right] + \beta\mathbb{E}_{p(\mathcal{G}, \mathcal{G}_s)}\left[\log \frac{p(\mathcal{G}, \mathcal{G}_s)}{p(\mathcal{G})p(\mathcal{G}_s)}\right] \\
&= -\mathbb{E}_{p(\hat{y}, \mathcal{G}_s)}\left[\log \frac{p(\hat{y}|\mathcal{G}_s)p(\mathcal{G}_s)}{p(\hat{y})p(\mathcal{G}_s)}\right] + \beta\mathbb{E}_{p(\mathcal{G}, \mathcal{G}_s)}\left[\log \frac{p(\mathcal{G}_s|\mathcal{G})p(\mathcal{G})}{p(\mathcal{G})p(\mathcal{G}_s)}\right] \\
&= -\mathbb{E}_{p(\hat{y}, \mathcal{G}_s)}\left[\log \frac{p(\hat{y}|\mathcal{G}_s)}{p(\hat{y})}\right] + \beta\mathbb{E}_{p(\mathcal{G}, \mathcal{G}_s)}\left[\log \frac{p(\mathcal{G}_s|\mathcal{G})}{p(\mathcal{G}_s)}\right] \\
&= -\mathbb{E}_{p(\mathcal{G}_s)}\mathbb{E}_{p(\hat{y}|\mathcal{G}_s)}\left[\log \frac{p(\hat{y}|\mathcal{G}_s)}{p(\hat{y})}\right] + \beta\mathbb{E}_{p(\mathcal{G})}\mathbb{E}_{p(\mathcal{G}_s|\mathcal{G})}\left[\log \frac{p(\mathcal{G}_s|\mathcal{G})}{p(\mathcal{G}_s)}\right].
\end{aligned}
\tag{40}
$$

Using Jensen's inequality and assuming that $p_f(\hat{y}|\mathcal{G}_s)$ is an approximation of $p(\hat{y}|\mathcal{G}_s)$, we can get:

$$
\begin{aligned}
-\mathbb{E}_{p(\mathcal{G}_s)}\mathbb{E}_{p(\hat{y}|\mathcal{G}_s)}\left[\log \frac{p(\hat{y}|\mathcal{G}_s)}{p(\hat{y})}\right] &\leq -\mathbb{E}_{p(\mathcal{G}_s)}\mathbb{E}_{p(\hat{y}|\mathcal{G}_s)}[\log p_f(\hat{y}|\mathcal{G}_s)] - \mathbb{E}_{p(\hat{y})}[\log p(\hat{y})] \\
&= -\mathbb{E}_{p(\mathcal{G}_s, \hat{y})}[\log p_f(\hat{y}|\mathcal{G}_s)] + \mathrm{H}(\hat{y}).
\end{aligned}
\tag{41}
$$

We introduce explain models:

$$
\begin{aligned}
&\beta\mathbb{E}_{p(\mathcal{G})}\mathbb{E}_{p(\mathcal{G}_s|\mathcal{G})}\left[\log \frac{p(\mathcal{G}_s|\mathcal{G})}{p(\mathcal{G}_s)}\right] \\
&= \mathbb{E}_{p(\mathcal{G})}\mathbb{E}_{p(\mathcal{G}_s|\mathcal{G})}\left[\log \frac{p_\alpha(\mathcal{G}_s|\mathcal{G})}{p(\mathcal{G}_s)} \cdot \frac{p(\mathcal{G}_s|\mathcal{G})}{p_\alpha(\mathcal{G}_s|\mathcal{G})}\right] \\
&= \mathbb{E}_{p(\mathcal{G})}\mathbb{E}_{p(\mathcal{G}_s|\mathcal{G})}\left[\log \frac{p_\alpha(\mathcal{G}_s|\mathcal{G})}{p(\mathcal{G}_s)}\right] + \mathbb{E}_{p(\mathcal{G})}\mathbb{E}_{p(\mathcal{G}_s|\mathcal{G})}\left[\log \frac{p(\mathcal{G}_s|\mathcal{G})}{p_\alpha(\mathcal{G}_s|\mathcal{G})}\right].
\end{aligned}
\tag{42}
$$

The second term is the KL divergence:

$$
\mathbb{E}_{p(\mathcal{G})}\mathbb{E}_{p(\mathcal{G}_s|\mathcal{G})}\left[\log \frac{p(\mathcal{G}_s|\mathcal{G})}{p_\alpha(\mathcal{G}_s|\mathcal{G})}\right] = \mathbb{E}_{p(\mathcal{G})}[\mathrm{KL}(p(\mathcal{G}_s|\mathcal{G})||p_\alpha(\mathcal{G}_s|\mathcal{G}))] \geq 0.
\tag{43}
$$

570 Therefore,

$$\mathrm{I}(\mathcal{G}; \mathcal{G}_s) \leq \mathbb{E}_{p(\mathcal{G})}\mathbb{E}_{p_\alpha(\mathcal{G}_s|\mathcal{G})}\left[\log\frac{p_\alpha(\mathcal{G}_s|\mathcal{G})}{q(\mathcal{G}_s)}\right] = \mathbb{E}_{p(\mathcal{G})}\left[\mathrm{KL}(p_\alpha(\mathcal{G}_s|\mathcal{G})\|q(\mathcal{G}_s))\right]. \qquad (44)$$

571 Combined with our previous derivation of the first term, we can get:

$$-\mathrm{I}(\hat{y}; \mathcal{G}_s) + \beta\,\mathrm{I}(\mathcal{G}; \mathcal{G}_s) \leq -\mathbb{E}_{p(\mathcal{G}_s,\hat{y})}\left[\log p_f(\hat{y}|\mathcal{G}_s)\right] + \mathrm{H}(\hat{y}) + \beta\mathbb{E}_{p(\mathcal{G})}\left[\mathrm{KL}(p_\alpha(\mathcal{G}_s|\mathcal{G})\|q(\mathcal{G}_s))\right]. \quad (45)$$

572 ## F Experiment Supplement

Table 5: Statistics of Datasets.

| Dataset | DBLP | ACM | Freebase |
|---------|------|-----|----------|
| Nodes | 26,128 | 10,942 | 43,854 |
| Edges | 239,566 | 547,872 | 151,034 |
| Node Types | 4 | 4 | 4 |
| Edge Types | 6 | 8 | 6 |
| Classes | 4 | 3 | 3 |

573 ### F.1 Datasets

574 We conduct experiments on three real-world datasets. According to the Heterogeneous Graph
575 Benchmark [24] settings, we randomly split the nodes with proportions of 24%, 6%, and 70% for
576 training, validation, and testing, respectively. The statistics of the three datasets are shown in Table 5.

577   • **DBLP**[1] is a computer science bibliography network that contains four types of nodes: Paper
578    (P), Author (A), Term (T), and Venue (V). The authors in this network are from four research
579    areas (*Database, Data Mining, Artificial Intelligence,* and *Information Retrieval*).

580   • **ACM**[2] is a citation network that contains four types of nodes: Paper (P), Author (A), Term
581    (T), and Subject (S). The papers in this network are divided into three classes (*Database,*
582    *Wireless Communication,* and *Data Mining*).

583   • **Freebase** [44] is a knowledge graph that contains four types of nodes: Movie (M), Actor
584    (A), Director (D) and Writer (W).

585 ### F.2 Baselines

586 Next, we provide details on the baselines used in our experiments.

587   • **PGExpaliner** [16] is a parameterized explainer that learns a mask for each edge to obtain
588    edge importance scores.

589   • **GNNExplainer** [32] maximizes the mutual information between the model's prediction on
590    the original input and the masked input by masking features and edges.

591   • **PGM-Explainer** [33] employs a Bayesian network-based approach, treating vertices in the
592    input graph as random variables to fit the GNN model's predicted label.

593   • **V-InfoR** [5] utilizes a parametric method, learning edge masks on the latent representations
594    to identify important edges.

595   • **AMExplainer** [36] optimizes GNN explanations by leveraging adversarial networks to
596    achieve both sparsity and prediction accuracy, ensuring compact and faithful explanation
597    sets.

598   • **Hete-PGE** is an extension of PGExplainer, where we replace the initial concatenation with
599    a relation-based attention learning module to enable learning of heterogeneous semantics.

600   • **xPath** provides fine-grained explanations by identifying causal nodes and their influence
601    paths through a novel graph rewiring algorithm, offering detailed insights into the model's
602    decision-making process.

---

[1]https://dblp.uni-trier.de

[2]http://dl.acm.org/

## F.3 Base Heterogeneous Graph Neural Network

Table 6: Node classification result using our heterogeneous Graph Neural Network.

| Dataset | DBLP | ACM | Freebase |
|---------|------|-----|----------|
| Micro-F1 | 92.64±0.14 | 92.32±0.12 | 68.99±0.20 |
| Macro-F1 | 92.16±0.19 | 92.40±0.11 | 63.57±0.36 |

In the experiment, we use a basic heterogeneous Graph Neural Network, which encodes the input graph through 2 layers of GCN, and then used a layer of attention learning module to learn different heterogeneous relations. For a heterogeneous graph, the feature spaces of different types of nodes are usually different. We use a mapping function to map the features of different types into a common feature space, as shown below:

$$\mathbf{z}_v = \mathbf{W}_m \mathbf{x}_v^A + \mathbf{b}_m, \tag{46}$$

where $A \in \mathcal{A}$ is the node type of node $v$, $\mathbf{W}_m$ is a learnable weight, and $\mathbf{b}_m$ is the bias. Then, in the shared space, we use GCN to obtain the node embeddings:

$$\mathbf{Z}^{(l)} = \text{GCN}(\mathbf{Z}^{(l-1)}, \mathbf{A}), \mathbf{Z}^{(0)} = \mathbf{Z}_v. \tag{47}$$

To learn the heterogeneous semantics of the heterogeneous graph, we introduce a type vector $\boldsymbol{\gamma}_v$ and learn relation information through an attention module:

$$\boldsymbol{\gamma}_i^q = \mathbf{W}_r^q \boldsymbol{\gamma}_i, \boldsymbol{\gamma}_j^k = \mathbf{W}_r^k \boldsymbol{\gamma}_j,$$
$$score_{ij}^{\boldsymbol{\gamma}} = \boldsymbol{\gamma}_i^q \boldsymbol{\gamma}_j^k, \tag{48}$$

where $\mathbf{W}_r^q$ and $\mathbf{W}_r^k$ are learnable weights. The attention of the nodes can be computed as follows:

$$q_i = \mathbf{W}_q^z \mathbf{z}_i, k_j = \mathbf{W}_k^z \mathbf{z}_j,$$
$$\widehat{\alpha}_{ij} = \frac{\exp(\text{LeakyReLU}(a^T[q_i \parallel k_j]))}{\sum_{j' \in \mathcal{N}_i} \exp(\text{LeakyReLU}(a^T[q_i \parallel k_{j'}]))}. \tag{49}$$

where $\mathbf{W}_q^z$ and $\mathbf{W}_k^z$ are learnable weights. The final prediction can be expressed as:

$$score_{ij} = \widehat{\alpha}_{ij} + \beta score_{ij}^{\boldsymbol{\gamma}},$$
$$\mathbf{Z}_{\mathbf{H}}^{(l)} = \text{LayerNorm}(\mathbf{Z}_{\mathbf{H}}^{(l-1)} + score_{ij} \cdot \mathbf{Z}_{\mathbf{H}}^{(l-1)}),$$
$$\hat{Y} = P_f(\mathbf{Z}_{\mathbf{H}}^{(l)}; \theta_p). \tag{50}$$

where $\theta_p$ is the parameter of the predictor. The basic prediction results are shown in Table 6.

The experiments are conducted on an L20 GPU with 48GB of memory. Our CPU is an Intel(R) Xeon(R) Platinum 8457C. We utilized PyTorch version 1.13.1 and DGL version 1.1.1.

## F.4 Impact of Noise

We measure the impact of noise on the graph distribution through the variation of the KL divergence. The result is in Table 7, and it can be observed that noise significantly disrupts the graph distribution.

Table 7: The impact of noise with different budgets on graph distribution.

| Noise | 10% | 20% | 30% | 40% | 50% | 60% | 70% | 80% | 90% |
|-------|-----|-----|-----|-----|-----|-----|-----|-----|-----|
| KL | 0.11108 | 0.31823 | 0.54854 | 0.80690 | 1.09782 | 1.43823 | 2.01607 | 2.15205 | 2.40943 |

## F.5 Parameter Setting

For the base heterogeneous Graph Neural Network, we use Adam [45] as the optimizer, LeakyReLU with a negative slope $s = 0.2$ as the activation function, a learning rate of 1e-4, and a dropout rate of 0 for Freebase and 0.5 for other datasets. The hidden dimension is set to 256. Our training is performed for 100 epochs.

## G Limitations

**Node Feature Noise.**   This paper focuses on structural noise. Both denoising variational inference and heterogeneous explanation generator are proposed to mitigate the impact of structural noise. Therefore, RoHeX cannot be directly used to explain heterogeneous graphs containing feature noise. We will extend RoHeX to achieve robustness at the node feature level in future work.

**Dynamic Graph.**   RoHeX can be used to explain different levels of tasks on heterogeneous graphs and can also be applied to homogeneous graphs. Due to the lack of dynamic information extraction module, RoHeX cannot be applied to dynamic graphs.

