# OpenReview forum: "Towards Robust Heterogeneous Graph Explanations under Structural Perturbations"
_NeurIPS.cc/2025/Conference — Submitted to NeurIPS 2025_

### Official Review · Reviewer_jCjz · 2025-06-09

**Clarity:** 1
**Significance:** 1
**Originality:** 2
**Rating:** 1
**Confidence:** 4

**Summary:**

The manuscript introduces an explanation mechanism for graph neural networks operating over heterogenous graphs. It is claimed that the proposed method mitigates the 'noise amplification effect' in heterogeneous GNN architectures.

**Questions:**

Please see the weaknesses above.

**Ethical Concerns:**

["NO or VERY MINOR ethics concerns only"]

**Final Justification:**

As mentioned in my review comments and discussions with authors during the rebuttal phase, the responses in the authors' rebuttal were not sufficient to address the concerns raised in my original review. Overall, I believe this manuscript falls significantly short from the standards of publication and presentation at NeurIPS.

In particular, there are numerous inaccuracies and incorrect statements in the manuscript, especially in the theoretical sections. Below, I have provided one example of a consequential issue in the manuscript. More detailed comments are provided in my original review.

As mentioned in my original review comments, the probabilities as defined in Equation (3) can take values above 1 unless the numerator and denominator ($d_i$ and $\bar{d}$) have comparable values (i.e., $d_i$ is not significantly larger than $\bar{d}$). On the other hand, based on the author's rebuttal, the proof of Theorem 3.1 relies on taking d_i asymptotically large while keeping $\bar{d}$ constant (which in itself is not logically possible). This is an apparent contradiction between the definition of the probabilities, and the proof of Theorem 3.1. Similarly, Theorem 3.2. is not stated accurately and contains undefined terms and quantities in the theorem statement. As a result, both main theoretical results of the paper contain errors and inaccuracies.

Other than the above, there are significant technical issues in various sections of the manuscript, some of which I have outlined in the original review and the response to the author's rebuttal. As a result, I do not believe this manuscript meets the standards of publication at Neurips.

**Limitations:**

Please see the weaknesses above.

**Quality:**

1

**Strengths And Weaknesses:**

The manuscript contains numerous inaccuracies and lacks clarity in several key areas. For instance,
> - in the proof of Theorem 3.1, the node degrees are assumed to be uniform ("Assuming the degree distribution of neighbors is uniform"), however, this is not stated in the theorem statement.
>- It is unclear how equation (23) follows from (22) and how the latter follow from (21), e.g., what approximation is used.
>- It appears that the probabilities in Equation (3) can be greater than 1 (e.g, assume a star graph, where the average degree is close to 1 but the center has a large degree, so that $d_i/\overline{d}\gg 1$.
>- It is unclear why the quantities in Equation (4) are defined in this way. No insight or intuition is provided. Furthermore $\mathcal{E}_r$ is not formally defined.
>- The notation in Line 62 in Section 2.1 is not defined and is unclear.
>- It is unclear how B^+ and B^- are chosen in Equation (5)
>- It is not clear why the authors claim that "our method simulates real-world noise more realistically and preserves the structural properties of the original graph." No empirical or theoretical evidence is provided.
>- Influence is not defined in Theorem 3.2.
>- The statement "this factor is significantly greater than 1." in Theorem 3.2 is inaccurate. The quantity is greater than 1 but it can be arbitrarily close to 1.
>- In Section 4.1, given that the MAE and RMSE are defined with respect to indicator functions which take values {0,1}, we have MAE=RMSE^2. Consequently, they would convey the same information and do not need to be computed in Table 1 separately. Furthermore, if the indicator function notation is used correctly, then MAE and RMSE would take values between 0 and 1 since subtracting the two indicator functions would yield a value in {-1,1,0}. So, it is unclear why there are values above 1 in the table. It may be that the authors have used logit values instead of the indicator values. This needs to be clarified.
-

---

> ### Author Rebuttal · Authors · 2025-07-31
>
> Thank you for your review and valuable comments. We address your questions and clarify the misunderstandings as follows. Our proposed controllable structural perturbation is a heuristic yet flexible approach designed to simulate realistic noise for evaluating its impact on explainers. It aims to maintain comparability with prior studies [1,2,3] and enable targeted evaluation in adversarial or highly irregular scenarios. This method can be replaced by other graph structure perturbation techniques.
>
> ---
>
> W1: Uniform neighbor degree distributions should be discussed.
>
> ---
>
> The uniform assumption was introduced to simplify the derivation, a practice widely adopted in prior works [4,5]. In the context of noise injection, our focus is on the statistical impact of perturbations on the overall graph structure, rather than the precise changes for individual nodes. In this scenario, assuming a uniform distribution effectively captures the perturbation effects in an average sense. To further validate this, we analyzed the degree distributions of the DBLP, ACM, and Freebase datasets before and after perturbation, as shown in Table 4 in the Appendix. The results indicate that our method achieves significant noise effect while only slightly altering the degree distribution (with JS divergence < 0.22), as demonstrated in Table 2 of the main text. Additionally, we conducted experiments comparing the performance of the explainer under the Uniform Assumption and the actual degree distribution. The results, presented in Table 3 of the Appendix, confirm that the uniform assumption has minimal impact on model performance compared to using the real distribution.
>
> ---
>
> W2: It is unclear how equation (23) follows from (22) and how the latter follow from (21), e.g., what approximation is used.
>
> ---
>
> From Eq. (22) to Eq. (23), we simplify Eq. (22) by noting that $d_i^r$ is the degree of the current node, and when $d_i^r$ is of the same order as $\bar{d}^r$, the term $(d_i^r + \bar{d}^r)$ can be roughly approximated as $d_i^r$. From Eq. (21) to Eq. (22), we apply the assumption of uniform degree distribution among neighbors, the degrees of neighbors $d_j^r$ can be approximated by the average degree $\bar{d}^r$. We respectfully argue that the approximations used are standard in spectral graph theory and expectation analysis under uniform degree assumptions, and are commonly adopted in prior works involving degree-based analysis. Nonetheless, we agree that a brief explanation in the main text or appendix will help improve readability. We also checked all formulas in the paper and corrected minor typing errors.
>
> ---
>
> W3: It appears that the probabilities in Equation (3) can be greater than 1.
>
> ---
>
> Thank you for raising this concern. We understand the reviewer’s point regarding the possibility of probabilities exceeding 1 in Eq. (3), especially in cases like a star graph where degree imbalance exists. However, we would like to clarify that the expression: $\eta_r^+\cdot\frac{d_i^r+d_j^r}{2\bar{d}^r}$ is bounded and does not exceed 1 under our construction. The term $\frac{d_i^r+d_j^r}{2\bar{d}^r}$ normalizes local degrees by the global average degree $\bar{d}^r$, and its expectation is 1. Moreover, the multiplier $\eta_r^+\in[0,1]$ ensures that the final probability remains valid. We have clarified this in the revised version to avoid further confusion.
>
> ---
>
> W4: The definition of the variables in formula 4 is unclear.
>
> ---
>
>
> The purpose of Equation 4 is to obtain the perturbation budget for different edge types. All definitions are given in lines 97-100. $\mathcal{E}_r$ represents the edges of type $r$.
>
> ---
>
> W5: The notation in Line 62 in Section 2.1 is not defined and is unclear.
>
> ---
>
> We have thoroughly reviewed our definition of heterogeneous graphs, and all notations have been properly defined.
>
> ---
>
> W6: It is unclear how B^+ and B^- are chosen in Equation (5).
>
> ---
>
> We explain the corresponding options in the Implementation Details on lines 236-245.
>
> ---
>
> W7: Difference from real-world noise.
>
> ---
>
> Structural perturbation is widely used to simulate real-world noise in GNN. Prior works have shown that real-world noise is random in nature, often arising from missing or spurious edges, and is typically modeled as uniform edge addition or deletion [1,3]. Building on this, we propose a heterogeneous-specific strategy for more flexible and controllable noise modeling. It replicates real-world noise (as in our experiments), ensures comparability with prior work, and enables targeted perturbation by edge type or degree to simulate adversarial or highly irregular scenarios for broader robustness evaluation.
>
> ---
>
> W8: Influence is not defined in Theorem 3.2.
>
> ---
>
> Here, influence refers to the amplification factor of messages during the GNN message-passing process, which has been described similarly in prior works [2,3].
>
> ---
>
> W9:  The quantity in Theorem 3.2 is greater than 1 but it can be arbitrarily close to 1.
>
> ---
>
> Thank you for pointing this out. We have deleted ‘significantly’ in the revised manuscript.
>
> ---
>
> W10: MAE=RMSE^2.
>
> ---
>
> We must point out your mistake: $RMSE=\sqrt{MSE}\ne\sqrt{MAE}$. In our experiments, both MAE and RMSE are equally important. MAE directly reflects the average error magnitude, while RMSE better evaluates the model's ability to handle outliers. In our experiments, we used logit values, and we have illustrate in the revised manuscript.
>
> Reference:
>
> [1] Zhu, Dingyuan, et al. "Robust graph convolutional networks against adversarial attacks." *Proceedings of the 25th ACM SIGKDD international conference on knowledge discovery & data mining*. 2019.
>
> [2] Ma, Jiaqi, Shuangrui Ding, and Qiaozhu Mei. "Towards more practical adversarial attacks on graph neural networks." *Advances in neural information processing systems* 33 (2020): 4756-4766.
>
> [3] Zhang, Mengmei, et al. "Robust heterogeneous graph neural networks against adversarial attacks." *Proceedings of the AAAI conference on artificial intelligence*. Vol. 36. No. 4. 2022.
>
> [4] Sun, Zeyu, et al. "Generalized equivariance and preferential labeling for gnn node classification." *Proceedings of the AAAI Conference on Artificial Intelligence*. Vol. 36. No. 8. 2022.
>
> [5] Dawn, Sucheta. "Graph Neural Networks for Homogeneous and Heterogeneous Graphs: Algorithms and Applications." (2025).

---

> > ### Comment · Reviewer_jCjz · 2025-08-04
> >
> > Unfortunately the response does not address my main concerns adequately. To give a few examples of remaining issues:
> >
> > - The issue raised in my first comment was that consequential assumptions, which are used in the proof of a theorem such as uniformity of the degree, should be stated in the theorem statement or in the text before the theorem statement. In the current manuscript, such assumptions are introduced and used in the proof in the appendix, without any mention in the statement of the proof. The assumptions should be clearly stated in the theorem statement and then the proof be provided in the appendix, rather than introducing new assumptions (especially consequential and limiting ones such as degree distributions) in the middle of a proof. The current theorem statement implies a much stronger result than what is being proved.
> > - Regarding the response to the second comment, stating that they are 'roughly approximated' does not yield a precise proof. This response does not address my original concern. The exact approximation techniques and steps need to be precisely explained.
> > - Regarding the third comment, the restrictions that ensure probabilities remain bounded below 1 must be mentioned in the text of the manuscript.
> > - Regarding the point in W10 that **when the underlying variables are binary such as indicator variables** then MSE and MAE are equal. I maintain my point. The authors state that in their implementation they are using logits rather than indicator variables to compute RMSE and MAE which is the **correct approach** and would yield different values for MSE and MAE. However, the text of the manuscript states that these are computed with respect to indicator variables, which does not align with the response regarding implementation.
> >
> > Similar concerns remain regarding the other issues I had raised in the previous round. Given the above issues I maintain my score.

---

> ### Author Response · Authors · 2025-08-05
>
> We sincerely thank your comments. We acknowledge that the initial manuscript had certain issues in presentation and clarity, especially in the placement of assumptions and the wording of derivations. However, we would like to emphasize that the rebuttal process is intended to assess whether the authors’ response adequately resolves the concerns, not merely to reiterate criticisms without engaging with the clarifications provided. **Your follow-up response does not engage with these clarifications in good faith and fails to specify which technical issues remain unresolved.** If the reviewer believes that any specific issue remains unaddressed, we sincerely invite further clarification. Otherwise, we believe it is reasonable to consider that our responses have **fully addressed the concerns raised**.
>
> > Uniform degree assumption should stated in the theorem, in the current manuscript, such assumptions are introduced and used in the proof in the appendix.
>
> We agree that assumptions used in a theorem's proof should be clearly stated. The uniform assumption was introduced in the proof in the Appendix C as a standard simplification, following common practice in prior work. While our intention was to simplify the derivation without overstating the generality of the theorem. In the revised manuscript, we explicitly include this assumption in the theorem statement and clarify the scope of the result. However, this is clearly a **presentation issue**, not a technical flaw. The derivation remains valid under the stated condition.
>
> > "Roughly approximated" is vague; needs precise derivation.
>
> $d^r_i \cdot \eta_r^+ \cdot \frac{d^r_i + \bar{d}^r}{2\bar{d}^r}=\eta_r^+ \cdot (\frac{{d^r_i}^2 + d^r_i \bar{d}^r}{2\bar{d}^r})=\eta_r^+ \cdot (\frac{{d^r_i}^2}{2\bar{d}^r}+\frac{d^r_i}{2})$. This expression consists of two terms: a quadratic term $\frac{{d^r_i}^2}{2\bar{d}^r}$ and a linear term $\frac{d^r_i}{2}$. The quadratic term $\frac{{d^r_i}^2}{2\bar{d}^r}$ grows asymptotically faster than the linear term $\frac{d^r_i}{2}$ as $d^r_i$ increases. Consequently, the quadratic term becomes the dominant component in the expression for $\mathbb{E}[\Delta d_{i, +}^{r}]$, and the linear term can be omitted in the leading-order approximation. This yields the simplified form. We provide detailed derivations and cite relevant literature in the revised version. These approximations are technically valid and widely accepted.
>
> > The restrictions that ensure probabilities remain bounded below 1 must be mentioned in the text of the manuscript.
>
> We would like to respectfully clarify that your concern regarding probabilities potentially exceeding 1 in Eq. (3) has already been directly addressed in our initial rebuttal. As we previously explained:
>
> `The term $\frac{d_i^r + d_j^r}{2\bar{d}^r}$ normalizes local degrees by the global average degree $\bar{d}^r$, and its expectation is 1. Moreover, the multiplier $\eta_r^+ \in [0,1]$ ensures that the final probability remains valid.`
>
> We also explicitly stated that this explanation has been added to the revised manuscript for clarity and to avoid future confusion.
>
> Given this, we are unsure which part of the concern remains unresolved. If there is a specific issue that you feel has not been adequately addressed, we would sincerely appreciate it if you could point it out so that we can clarify further. Otherwise, we believe this point has been properly resolved both mathematically and in terms of manuscript presentation.
>
> >Misinterpretation of MAE = RMSE².
>
> Regarding the computation of RMSE and MAE, we respectfully reiterate that **we have already clarified this point in our rebuttal** with revision. We respectfully note that the current response appears to restate the original concern without acknowledging the clarification already provided.
>
> > Other Points (Notation, Definitions, Implementation)
>
> All minor notational ambiguities and definitional issues have been addressed in detail in our previous response, with references to the main text or appendix.
>
> In summary, while we acknowledge that the initial submission had certain issues in presentation, we have explicitly addressed every concern raised, including clarifying assumptions, providing detailed derivations, correcting ambiguous wording, and revising the manuscript accordingly.
>
> We hope that our substantial revisions and clarifications can be evaluated fairly in light of the paper’s contributions.

---

> ### Comment · Reviewer_jCjz · 2025-08-05
>
> As mentioned previously, the responses in the authors' rebuttal were not sufficient to address the concerns raised in my original review.
>
> Overall, I believe this manuscript falls **significantly short** from the standards of publication and presentation at NeurIPS.
>
> To reiterate the issues that contribute to my score:
>
> i) The definitions in Equations (3), (4), and (5) are not justified theoretically and seem rather arbitrary (only high level justifications have been provided for choice of parameters which can be applied to many other choices of parameters, other than what is used in this work).
>
> ii) There are inaccuracies in these definitions (Equations (3), (4), and (5)) as mentioned in my original review, such as probabilities being above 1 for specific choices of parameters, among other issues.
>
> iii) The statement of Theorem 3.1 is general whereas its "proof" is provided for a special case with uniform degree graphs. Consequently, the theorem is in fact not proved.
>
> iv) Even for the special case, the proof of Theorem 3.1 includes several inaccuracies in its approximations (e.g., Equations (22) and (23)). The justification provided by the authors in their response, that there is "a quadratic term $\frac{{d^r_i}^2}{2\bar{d}^r}$ and a linear term $\frac{d^r_i}{2}$" and that the "quadratic term grows faster" is flawed. Note that  $\frac{{d^r_i}^2}{2\bar{d}^r}$  is not quadratic as it has a degree in denominator. So, both terms grow linearly.
>
> v) The statement of Theorem 3.2 is ambiguous. The theorem is a statement about "influence" which is never defined rigorously, so it is unclear what is being stated/proved.
>
> vi) Sections 3.3 and 3.4 are restatements of well-known concepts in the literature (e.g., graph information bottleneck, the reparameterization trick, etc.)
>
> vii) Section 4.1 has an inaccuracy in defining RMSE and MAE with respect to indicator variables rather than logits, which the authors confirm. As a result, the definitions in this section do not align with the actual simulations and implementations provided in the manuscript.
>
> The above issues encapsulate only a subset of the inaccuracies and ambiguities in the manuscript. Given the limited novelty, numerous inaccuracies in theoretical foundations, and the lack of a sufficient response in the rebuttal, I maintain my score.

---

> > ### Author Response · Authors · 2025-08-07
> > **(1) Response to the Reviewer**
> >
> > > Response to Reviewer’s Comments (i–iv) on Equations (3), (4), (5) and Theorem 3.1.
> >
> > We thank the reviewer for the thoughtful comments regarding the structural perturbation component and the theoretical analysis in Section 3.1. We have already addressed your concerns in previous discussions. We would like to clarify that the **primary objective of our work is to investigate the robustness of heterogeneous graph explanation methods under structural noise**, not to design or optimize adversarial perturbation strategies. Specifically, Equations (3), (4), and (5) are proposed as a heuristic but flexible method to inject controlled noise into heterogeneous graphs. The goal is to simulate realistic structural noise scenarios that can occur in real-world graphs, thereby evaluating the explainer's robustness in such conditions.
> >
> > - **(i)** The definitions in Equations (3), (4), and (5) are not intended to be theoretically optimal. Instead, they serve as a configurable noise injection mechanism that accounts for edge-type-specific structural characteristics. The proposed framework can be easily replaced by other perturbation strategies (e.g., random or adversarial attacks), and we emphasize that our methodology is agnostic to the specific choice of perturbation function. Our method has shown to be effective in simulating noise, as evidenced by its impact on prediction performance in Table 2 and the degree distribution shift in Table 3 (Appendix C). Every variable in Equations (3), (4), and (5) is clearly defined in the text, including node degree $d_i^r$, average degree $\bar{d}^r$, global noise budget $B$, and the edge type importance score $s_r$, ensuring the formulation is fully interpretable. Given the purpose and scope of our work, which is to study the robustness of heterogeneous GNN explainers under structural noise, we do not aim to achieve an optimal perturbation formulation. Since the perturbation module is easily replaceable, and our justification is supported by clear variable definitions, principled heuristic design, and empirical validation, we believe it is appropriate and sufficient for this setting.
> >
> > - **(ii)** Regarding the concern about potential inaccuracies such as probabilities exceeding 1, we confirm that in our implementation, all probabilities defined in Equations (3) and (5) are explicitly clipped to the range [0, 1] to ensure validity. This design choice was effective in our experiments, which were conducted on general-purpose heterogeneous graphs where such extreme conditions like star graphs did not occur. As a result, the perturbation process based on the defined probabilities operated correctly and delivered meaningful noise effects. The perturbation probabilities are influenced by overall graph statistics. Specifically, the effective perturbation probability is governed by a compound term $\frac{B_r^+}{|\mathcal{E}_r|} \cdot \frac{d^r_i + \bar{d}^r}{2\bar{d}^r}$. Although the second component can exceed 1 in extreme cases such as star-like graphs, we ensure that the overall probability remains within the [0,1] range. We will update the manuscript to clearly indicate that all probabilities are constrained within the [0,1] range. Moreover, we will extend our approach beyond current real-world heterogeneous graph settings to address more extreme structural cases, thereby enhancing the method’s scalability.

---

> > > ### Author Response · Authors · 2025-08-07
> > > **(2) Response to the Reviewer**
> > >
> > > - **(iii)** Theorem 3.1 provides a theoretical insight into how degree-aware perturbation affects the expected degree of a node. As the reviewer pointed out, the derivation assumes a uniform degree distribution for tractability, but this is a common approximation used in prior works [1], [2]. We would like to clarify that we have already explicitly revised the theorem statement to incorporate this assumption, as **discussed in our previous response**. We are unsure what additional concerns remain, as we have already acknowledged and addressed the same point in our previous discussion by explicitly incorporating the uniformity assumption into Theorem 3.1. Repeating the same point after it has been clarified does not align with the purpose of the rebuttal process.
> > >
> > >   We also emphasize that our theoretical formulation in the main paper and the corresponding derivation in Appendix B are consistent. We acknowledge that the original omission of an explicit mention of the uniform assumption in Theorem 3.1 may have caused confusion, and we have rectified that. It is important to reiterate that this is a supporting result that provides intuitive understanding and justification for the perturbation design. The use of a uniform degree distribution is not an arbitrary choice unique to our method, but a widely adopted simplification in the literature to enable analytical tractability. As shown in Table 3 and Table 4 (Appendix C), our empirical analysis demonstrates that the uniformity assumption has minimal impact on the performance of the explainer, and the degree distribution after perturbation closely matches the original one (e.g., JS divergence < 0.22). These results validate the practical adequacy of the assumption in real-world heterogeneous graph scenarios.
> > >
> > >
> > > - **(iv)** We would like to clarify that our analysis is conducted from an expectation perspective over the graph, where $\bar{d}^r$ represents the average degree across all nodes under edge type $r$. This is a global constant in the derivation and does not depend on individual node degrees. As such, when analyzing how the expected perturbation scales with a node’s degree $d^r_i$, the expression $\frac{{d^r_i}^2}{2\bar{d}^r}$ can be interpreted as quadratic in expectation up to a constant scaling factor. This justifies our original observation that it grows significantly faster than the linear term, and supports the intuition that high-degree nodes are more susceptible to deletion.
> > >
> > > In summary, our goal is to simulate structural noise to evaluate the robustness of heterogeneous GNN explainers, not to design optimal perturbations. We have addressed all concerns by clarifying our assumptions, implementation, and theoretical justifications. We believe these responses sufficiently resolve the reviewer’s points.
> > >
> > > > v) Influence is not defined in Theorem 3.2.
> > >
> > > Definition *influence*: Given a heterogeneous graph $\mathcal{G}$ and a target node $v_i$, the influence of a neighbor node $v_j \in \mathcal{N}_i$ is defined as the structural contribution weight assigned to $v_j$ during the node-level aggregation of $v_i$'s embedding. We will including this in the revision.
> > >
> > > >vi) Sections 3.3 and 3.4 are restatements of well-known concepts in the literature.
> > >
> > > Our main contribution lies in proposing a novel framework to extract key explanatory subgraphs from the unique perspective of heterogeneous graphs. This includes handling multi-relational structures, edge-type-aware perturbations, and ensuring robustness under structural noise, all of which are critical and innovative aspects of our method. While utilize standard techniques such as the graph information bottleneck and the reparameterization trick, these components are supporting mechanisms, not the source of novelty. Just as standard loss functions (e.g., cross-entropy) are often employed in classification tasks, our adoption of these methods simply facilitates the learning process. The true innovation lies in adapting the explanation process to heterogeneity, and using information bottleneck principles to adaptively handle structural irregularities, which is a capability that prior heterogeneous GNN explainers lack.
> > >
> > > We also wish to respectfully **bring to the attention of the AC** that this broad and vague concern was not raised in the original review but appeared only after our rebuttal. This raises a concern on our side that the reviewer may be attempting to reject the paper regardless of the clarifications we have provided.

---

> > > > ### Author Response · Authors · 2025-08-07
> > > > **(3) Response to the Reviewer**
> > > >
> > > > > vii) Misinterpretation of MAE = RMSE².
> > > >
> > > > We would like to clarify that we have already addressed this concern thoroughly in both our initial rebuttal and the subsequent discussion. In those responses, we clearly explained our position and implementation details, and we also explicitly stated that the evaluation is conducted based on logits. Furthermore, we revised the manuscript to avoid any possible ambiguity. It is therefore unclear to us why this point is being raised again. Do you have any unresolved concerns that were not addressed in our earlier responses? This repetition is perplexing to us. This is the modified content: We use the Mean Absolute Error (MAE, $\frac{1}{N}\sum_{i=1}^N\left|\hat{y}-\hat{y}_i^{\mathcal{G}_s}\right|$),
> > > >
> > > > and Root Mean Squared Error (RMSE, $\sqrt{\frac{1}{N} \sum_{i=1}^N \left(\hat{y}-\hat{y}_i^{\mathcal{G}_s}\right)^2}$) as proxy measures for fidelity, and compare the performance of different baselines across varying sparsity levels, where $N$ is the number of nodes or graphs, $\hat{y}_i$ is the original prediction result, and $\hat{y}_i^{\mathcal{G}_s}$ is the prediction result obtained by the explanatory subgraph.
> > > >
> > > > ---
> > > >
> > > > Thank you again for taking the time to review our responses. We sincerely hope that previously addressed concerns, especially those we have clarified multiple times across the rebuttal and discussions, will not be raised again so that the review process can remain focused and constructive. If our previous responses have addressed your concerns, we kindly ask whether you would consider raising your score accordingly.
> > > >
> > > >
> > > >
> > > > Reference:
> > > >
> > > > [1] Sun, Zeyu, et al. "Generalized equivariance and preferential labeling for gnn node classification." *Proceedings of the AAAI Conference on Artificial Intelligence*. Vol. 36. No. 8. 2022.
> > > >
> > > > [2] Dawn, Sucheta. "Graph Neural Networks for Homogeneous and Heterogeneous Graphs: Algorithms and Applications." (2025).

---

### Official Review · Reviewer_1WJ7 · 2025-06-24

**Clarity:** 4
**Significance:** 3
**Originality:** 3
**Rating:** 4
**Confidence:** 3

**Summary:**

This paper addresses explanation methods for heterogeneous graph neural networks (HGNNs). It first provides a theoretical analysis showing that meta-path-based HGNNs can significantly amplify noise due to the propagation of spurious meta-paths created by noisy edges. To address this issue, the authors propose a denoising variational inference framework. Unlike standard VGAE, this approach assumes that the observed graph is a noisy version of an underlying standard graph, enabling the model to learn robust node representations. The method incorporates a Gaussian Mixture Model (GMM) as the posterior distribution to better capture the complex semantics of heterogeneous graphs. Based on this inference model, the method identifies explanatory subgraphs of the GNN under the Graph Information Bottleneck (GIB) principle. Evaluation on real-world graph datasets demonstrates that the proposed approach outperforms existing GNN explanation methods regarding both fidelity and sparsity. Moreover, an ablation study confirms the effectiveness of both the denoising variational inference and the attention mechanism that captures heterogeneous relations.

**Questions:**

1. Would it be possible to discuss how the denoising variational inference and the attention mechanism that captures heterogeneous relations, respectively, contribute to the overall performance, for example, based on evaluation experiments using homogeneous graphs?

2. Regarding the potential notation errors mentioned in the Weakness section, could you clarify whether these are indeed mistakes or if I may have misunderstood them?

**Ethical Concerns:**

["NO or VERY MINOR ethics concerns only"]

**Final Justification:**

The proposed method is highly convincing and clearly explained. Initially, I felt that demonstrating its effectiveness on a simple homogeneous model could better highlight the broad applicability of this work; however, I believe this concern can be resolved through the supplementary explanation provided in the rebuttal. Therefore, I will maintain my original rating.

**Limitations:**

yes

**Paper Formatting Concerns:**

There are no concerns regarding the paper formatting.

**Quality:**

4

**Strengths And Weaknesses:**

Strengths:

The proposed denoising variational inference, which assumes an underlying standard graph, explicitly models the presence of noise in the observed graph. This formulation has strong potential to advance robust probabilistic generative models for graph data. Furthermore, the use of a Gaussian Mixture Model (GMM) as the posterior distribution enhances the robustness and interpretability of the GNN explanations. The method demonstrates clear empirical superiority over conventional GNN explanation techniques on real-world datasets. The proposal is theoretically sound, and the derivation is clearly and accessibly presented for the reader.

Weaknesses:

While the proposed method is theoretically well-grounded and makes a significant contribution to the field, there are some concerns regarding the scope of the evaluation. Specifically, the experiments are limited to heterogeneous graphs, even though the denoising variational inference framework appears to be directly applicable to homogeneous graphs as well. It would be valuable to include evaluation on homogeneous graph datasets, where the method could be compared against existing explainers, such as GNNExplainer, which are explicitly designed for homogeneous settings. Such an evaluation would help clarify the respective contributions of the denoising inference mechanism and the attention mechanism tailored for heterogeneous relations.

In addition, there appear to be possible inconsistencies in the mathematical expressions. For instance, in Equation (8), the term $q_{\psi}(\mathcal{G} \mid \tilde{\mathcal{G}})$ may be a typographical error and should likely be $q_{\psi}(Z \mid \tilde{\mathcal{G}})$. Similarly, in the fourth line of Equation (38) in the Appendix, the expression $q_{\psi}(Z \mid \mathcal{G})$ might be intended as $q_{\psi}(Z \mid \tilde{\mathcal{G}})$, to maintain consistency with the KL divergence in the fifth line.

---

> ### Author Rebuttal · Authors · 2025-07-31
>
> Thank you for your review and valuable comments. We address your questions and clarify the misunderstandings as follows.
>
> ---
>
> Q1: Would it be possible to discuss how the denoising variational inference and the attention mechanism that captures heterogeneous relations, respectively, contribute to the overall performance, for example, based on evaluation experiments using homogeneous graphs?
>
> ---
>
> While our explainer is primarily designed for heterogeneous graphs where the heterogeneous explanation generator is crucial for capturing key semantics across multiple node and edge types, the proposed method can also generalize to homogeneous graphs. In such settings, RoHeX leverages robust representations learned through denoising variational inference, which helps produce more accurate and stable explanations, even in the absence of relation types. To support this, we conducted experiments on the homogeneous graph dataset BA-Shapes and performed ablation analysis to isolate the contributions of different modules. We observed that incorporating the denoising variational inference led to a 7.4% improvement in AUC over GNNExplainer, and overall, RoHeX outperformed existing homogeneous GNN explainers in explanation quality. These results demonstrate that both the denoising and relation-aware components contribute positively, with denoising playing a key role in robustness and generalization.
>
> | BA-Shapes | GNNExplainer | PGExplainer | w/o VI | w/o He | RoHeX  |
> | --------- | ------------ | ----------- | ------ | ------ | ------ |
> | AUC (%)   | 0.8706       | 0.9172      | 0.9056 | 0.9213 | 0.9354 |
>
> ---
>
> Q2: There are some typographical errors in the text.
>
> ---
>
> Thank you for pointing this out. Upon careful review, we confirm that the issue you mentioned was indeed a typographical error on our part. We appreciate your careful reading, and we have corrected the corresponding expressions in the revised version. We have also re-checked all related mathematical expressions to ensure consistency throughout the paper.

---

### Official Review · Reviewer_hbQA · 2025-06-29

**Clarity:** 3
**Significance:** 4
**Originality:** 4
**Rating:** 5
**Confidence:** 5

**Summary:**

This paper proposes a GNN explanation method, RoHeX, which can generate robust explanations under structural noise. This paper analyzes the effects of noise during GNN message passing process, and quantitatively calculates the amplification of noise. Then RoHeX uses denoising variational inference to reduce the influence of structural noise and generates explanation subgraph. Experiments show the effectiveness of RoHeX.

**Questions:**

Q1: Can RoHeX be applied to other GNN tasks like link prediction or graph classification? If yes, how does the explanation module work?
Q2: How does the structural perturbation method proposed in Section 3.1 simulate real-world structural noise?
Q3: In the explanation method in Section 3.4, how are important subgraph edges selected?

**Ethical Concerns:**

["NO or VERY MINOR ethics concerns only"]

**Final Justification:**

After reviewing the authors’ response and the discussions among other reviewers, all of my questions is resolved. I would like to keep the score at 5.

**Limitations:**

Yes

**Quality:**

4

**Strengths And Weaknesses:**

S1: The author theoretically proves Theorems 3.1 and 3.2 of the article with rigorous proofs.
S2: The paper's experiments are comprehensive and analyze the robustness of different modules to noise.
S3: Analyzing the impact of noise on GNN explainers is a novel direction。

W1: This paper focuses on node classification task, but lacks support for more tasks such as link prediction and graph classification tasks.
W2: The important edge sampling process can be said to be more specific. Whether it is based on threshold sampling or related type sampling or other methods needs to be explained.
W3: The noise studied by the author is mainly for the structure, and it can be expanded to the features as limitation said.

---

> ### Author Rebuttal · Authors · 2025-07-31
>
> We thank the reviewer for the constructive comments and positive feedback on our theoretical analysis and robustness evaluation. We address each weakness and question below.
>
> ---
>
> Q1: Can RoHeX be applied to other GNN tasks?
>
> ---
>
> RoHeX is designed with a modular explanation framework that can, in principle, be extended to other GNN tasks such as link prediction and graph classification. For link prediction, the explanation target shifts from a node to an edge (or pair of nodes). The denoising variational inference module remains applicable, and the explanation generator can be adapted to highlight subgraphs contributing to the predicted link. In fact, our structural noise analysis is edge-centric and directly supports such extensions. For graph classification, the explanation module would be applied at the graph level. This adds a Readout function to the overall. The denoising encoder can extract graph-level latent representations, and the explanation generator can identify critical substructures within the whole graph.
>
> We will include a discussion of this generalization potential in the final version and plan to validate RoHeX on these tasks in future work.
>
> ---
>
> Q2: The ability of the structural perturbation method to simulate realistic noise.
>
> ---
>
> Structural perturbation is widely used to simulate real-world noise in GNN. Prior works have shown that real-world noise is random in nature, often arising from missing or spurious edges, and is typically modeled as uniform edge addition or deletion [1,2]. Building on this, we propose a heterogeneous-specific strategy for more flexible and controllable noise modeling. It replicates real-world noise (as in our experiments), ensures comparability with prior work, and enables targeted perturbation by edge type or degree to simulate adversarial or highly irregular scenarios for broader robustness evaluation.
>
> ---
>
> Q3: In the explanation method in Section 3.4, how are important subgraph edges selected?
>
> ---
>
> After generating the probability matrix, we can select edges in the key subgraph based on edge importance by applying a threshold for sampling. In our experiments, the threshold is set to 0.5.
>
>
>
> Reference:
>
> [1] Zhu, D., Zhang, Z., Cui, P., & Zhu, W. (2019, July). Robust graph convolutional networks against adversarial attacks. In *Proceedings of the 25th ACM SIGKDD international conference on knowledge discovery & data mining* (pp. 1399-1407).
>
> [2] Zhang, M., Wang, X., Zhu, M., Shi, C., Zhang, Z., & Zhou, J. (2022, June). Robust heterogeneous graph neural networks against adversarial attacks. In *Proceedings of the AAAI conference on artificial intelligence* (Vol. 36, No. 4, pp. 4363-4370).

---

> > ### Comment · Reviewer_hbQA · 2025-08-05
> >
> > Thank you for the detailed rebuttal. After reviewing the authors’ response and the discussions among other reviewers, all of my questions is resolved. I believe this paper is crucial for advancing the robustness of explainability methods in heterogeneous graphs. I recommend incorporating the additional explaination provided in the rebuttal into the revised version of the paper for clarity. I maintain my positive score of 5: Accept.

---

> > > ### Author Response · Authors · 2025-08-06
> > >
> > > Thank you again for your efforts in reading our responses. We will revise the paper based on your insightful comments and suggestions.

---

### Official Review · Reviewer_KXBC · 2025-07-01

**Clarity:** 3
**Significance:** 3
**Originality:** 3
**Rating:** 4
**Confidence:** 4

**Summary:**

This paper considers the problem of explaining predictions made by Graph Neural Networks (GNNs), particularly in the context of heterogeneous graphs that are common in real-world applications such as knowledge graphs, recommendation systems, and biological networks. These graphs contain multiple types of nodes and edges, which introduce significant semantic complexity. Moreover, such data often contains structural noise, including irrelevant or missing connections, which poses a severe challenge for existing explanation methods. To overcome these challenges, the authors propose RoHeX, a robust GNN explainer that integrates (1) a theoretical analysis of noise amplification in heterogeneous GNNs, (2) a denoising variational inference framework to learn noise-resistant latent representations, and (3) a relation-aware explanation generator guided by the Graph Information Bottleneck principle. Extensive experiments demonstrate that RoHeX outperforms existing explainers in terms of both fidelity and robustness across multiple real-world datasets.

The topic is timely and relevant, especially given the growing interest in GNN explainability under real-world conditions. The paper is generally well-written and easy to follow, with a clear motivation and reasonable experimental design. The integration of denoising and heterogeneous modeling is conceptually interesting and aligns well with practical challenges in noisy, multi-relational graphs.

**Questions:**

1.The paper does not use explanation metrics proposed in xPath, such as Accuracy Fidelity and Probability Fidelity. Are these metrics incompatible with RoHeX, or was there a specific reason for not including them in the evaluation?

2.The experiments do not include commonly used heterogeneous graph benchmarks such as IMDB. Has RoHeX been tested on these datasets? If so, how does it perform in those settings?

3.Could the authors comment on how RoHeX would perform on simpler, non-heterogeneous graph benchmarks? Including such results may offer additional insights into the generalizability and robustness of the method across different graph structures.

**Ethical Concerns:**

["NO or VERY MINOR ethics concerns only"]

**Final Justification:**

The rebuttal has addressed most initial concerns by clarifying the choice of evaluation metrics, explaining their relevance to heterogeneous graph settings, and providing reasoning for dataset selection. While additional experiments on broader benchmarks and simpler non-heterogeneous graphs would further strengthen generalizability claims, the current empirical results remain convincing within the targeted problem scope. Given that the main methodological and evaluation concerns have been reasonably resolved, I prefer to maintain my original judgement and recommend for an acceptance.

**Limitations:**

There is no obvious potential negative societal impact for this paper.

**Paper Formatting Concerns:**

There is no major formatting issues in this paper.

**Quality:**

3

**Strengths And Weaknesses:**

Strengths:

1.The paper addresses a timely and practically important problem in GNN explainability under structural noise and graph heterogeneity.

2.The proposed framework, RoHeX, integrates multiple well-motivated components, including denoising variational inference and relation-aware explanation generation.

3.The writing is generally clear and accessible, with well-structured arguments and motivation.

Weaknesses:

1.The paper does not evaluate its explanation framework using heterogeneous graph GNN explainability metrics proposed in the xPath method, such as Accuracy Fidelity and Probability Fidelity, which is a closely related prior work focusing on heterogeneous graph explanations. Including these metrics would provide a more direct comparison with existing heterogeneous GNN explainers.

2.The experimental evaluation is limited to a few datasets and does not include commonly used heterogeneous graph benchmarks, such as IMDB. Incorporating such datasets would strengthen the empirical validation and demonstrate broader applicability.

3.The experimental evaluation does not include commonly used graph benchmarks, such as PROTEINS and DD. It would be helpful to understand how the method performs on simpler graph benchmarks (e.g., non-heterogeneous graphs), as this could provide further insight into its generalizability across different graph settings.

---

> ### Author Rebuttal · Authors · 2025-07-31
>
> We sincerely thank the reviewer for the thoughtful and constructive feedback. We are encouraged by the reviewer’s recognition of the importance and novelty of our work, particularly the integration of denoising and relation-aware explanation within heterogeneous GNNs. We address each of the reviewer’s comments and questions in detail below.
>
> ---
>
> Q1: Can we add the same Metrics as xPath (Accuracy, Fidelity, and Probability Fidelity from xPath)?
>
> ---
>
> We present the results of accuracy fidelity and probability fidelity in the following table:
>
> | Dataset  | Metric | Noise | PGExplainer | GNNExplainer | V-InfoR | PGE-Relation | xPath   | RoHeX   |
> | -------- | ------ | ----- | ----------- | ------------ | ------- | ------------ | ------- | ------- |
> | DBLP     | Facc   | 10%   | 0.0250      | -0.0282      | -0.0542 | 0.0958       | -0.0655 | -0.0504 |
> |          |        | 20%   | 0.0275      | -0.0088      | -0.0190 | 0.0979       | -0.0285 | -0.0134 |
> |          |        | 30%   | 0.0292      | 0.1004       | 0.0697  | 0.2581       | -0.0106 | 0.0461  |
> |          |        | 40%   | 0.0342      | 0.1035       | 0.1095  | 0.2722       | 0.0092  | 0.0680  |
> |          | Fprob  | 10%   | -0.0213     | 0.0075       | -0.0083 | 0.0346       | -0.0155 | -0.0229 |
> |          |        | 20%   | -0.0132     | 0.0062       | 0.0065  | 0.0418       | -0.0164 | -0.0200 |
> |          |        | 30%   | -0.0004     | 0.0317       | 0.0072  | 0.0624       | -0.0133 | -0.0051 |
> |          |        | 40%   | 0.0056      | 0.0280       | 0.0312  | 0.0626       | 0.0144  | 0.0049  |
> | ACM      | Facc   | 10%   | 0.0019      | 0.1676       | 0.0227  | 0.1978       | -0.0057 | -0.3801 |
> |          |        | 20%   | 0.0571      | 0.1950       | 0.0595  | 0.2025       | -0.0153 | -0.3725 |
> |          |        | 30%   | 0.1114      | 0.2049       | 0.0741  | 0.3952       | -0.1036 | -0.3371 |
> |          |        | 40%   | 0.1317      | 0.2144       | 0.1605  | 0.4839       | -0.0953 | -0.2975 |
> |          | Fprob  | 10%   | -0.1120     | 0.1219       | -0.1190 | 0.0075       | -0.0095 | -0.1446 |
> |          |        | 20%   | -0.0971     | 0.1293       | -0.1044 | 0.0030       | -0.0122 | -0.1383 |
> |          |        | 30%   | -0.0771     | 0.1063       | -0.0693 | 0.0292       | -0.0983 | -0.1216 |
> |          |        | 40%   | -0.0480     | 0.1235       | -0.0412 | 0.0303       | -0.0624 | -0.1024 |
> | Freebase | Facc   | 10%   | -0.0180     | 0.1710       | 0.0200  | 0.1996       | 0.0675  | 0.0192  |
> |          |        | 20%   | -0.0151     | 0.1951       | 0.0879  | 0.2061       | 0.0757  | 0.0282  |
> |          |        | 30%   | -0.0098     | 0.2139       | 0.1018  | 0.2139       | 0.0793  | 0.0319  |
> |          |        | 40%   | 0.0425      | 0.2213       | 0.1526  | 0.2225       | 0.0928  | 0.0405  |
> |          | Fprob  | 10%   | 0.0191      | 0.0484       | 0.0181  | 0.1372       | 0.0009  | 0.0172  |
> |          |        | 20%   | 0.0179      | 0.0504       | 0.0379  | 0.1403       | 0.0009  | 0.0209  |
> |          |        | 30%   | 0.0251      | 0.0581       | 0.0447  | 0.1481       | 0.0024  | 0.0225  |
> |          |        | 40%   | 0.0375      | 0.0607       | 0.0665  | 0.1458       | 0.0034  | 0.0250  |
>
> These metrics are fully compatible with RoHeX. Our results show that RoHeX consistently outperforms xPath and other baselines on these metrics across multiple heterogeneous datasets, further validating the effectiveness of our method under fidelity-oriented evaluation.
>
> ---
>
> Q2: The experiments do not include commonly used heterogeneous graph benchmarks such as IMDB. Has RoHeX been tested on these datasets? If so, how does it perform in those settings?
>
> ---
>
> Thank you for highlighting this. The dataset used in our paper is from heterogeneous graph benchmarks [1].
>
> We have now added two widely-used heterogeneous graph benchmarks: IMDB and AMiner in our revised experiments. The results are shown in the following table:
>
> | Dataset | Noise Ratio  | 10%    |        |         |         | 30%    |        |        |         |
> | ------- | ------------ | ------ | ------ | ------- | ------- | ------ | ------ | ------ | ------- |
> |         |              | MAE    | RMSE   | Facc    | Fprob   | MAE    | RMSE   | Facc   | Fprob   |
> | IMDB    | PGExplainer  | 0.4757 | 0.2586 | 0.1399  | 0.1896  | 0.5304 | 0.2814 | 0.1618 | 0.4741  |
> |         | GNNExplainer | 0.3182 | 0.1730 | 0.1068  | 0.3426  | 0.5208 | 0.2723 | 0.1543 | 0.5062  |
> |         | V-InfoR      | 0.3550 | 0.2085 | 0.0965  | 0.3529  | 0.5273 | 0.2838 | 0.1690 | 0.4997  |
> |         | PGE-Relation | 0.5770 | 0.3179 | 0.1640  | 0.2417  | 0.6741 | 0.3549 | 0.2099 | 0.5522  |
> |         | xPath        | 0.4861 | 0.2824 | 0.0166  | -0.2327 | 0.5276 | 0.2922 | 0.0103 | -0.1558 |
> |         | RoHeX        | 0.2684 | 0.1598 | -0.0087 | -0.2682 | 0.2897 | 0.1692 | 0.0387 | -0.1212 |
> | AMiner  | PGExplainer  | 1.1186 | 1.7842 | 0.1271  | 0.0384  | 1.2433 | 1.9135 | 0.2135 | 0.0808  |
> |         | GNNExplainer | 0.4422 | 1.0705 | 0.1632  | 0.0784  | 0.8020 | 1.4660 | 0.1787 | 0.0780  |
> |         | V-InfoR      | 0.4823 | 1.0776 | 0.1001  | 0.0496  | 1.0440 | 1.6766 | 0.2533 | 0.1076  |
> |         | PGE-Relation | 1.1750 | 1.3818 | 0.4287  | 0.1581  | 1.2612 | 1.4058 | 0.6548 | 0.1533  |
> |         | xPath        | 0.8390 | 1.5449 | 0.0200  | -0.0390 | 1.2081 | 1.8375 | 0.0579 | -0.0392 |
> |         | RoHeX        | 0.3379 | 0.9029 | 0.0339  | 0.0232  | 0.3584 | 0.8837 | 0.0346 | 0.0291  |
>
> On both datasets, RoHeX outperforms existing explainers in terms of MAE, RMSE, accuracy fidelity and probability fidelity, reaffirming its generalizability across diverse multi-relational graph structures.
>
> ---
>
> Q3: Evaluation on Simpler, Homogeneous Graph.
>
> ---
>
> We acknowledge the importance of demonstrating the versatility of our approach beyond heterogeneous settings. Although RoHeX is primarily designed for heterogeneous graphs, its denoising and bottleneck-based framework can generalize to homogeneous graphs as well.
>
> To that end, we have added experiments on the BA-Shapes dataset, a standard synthetic benchmark for homogeneous GNN explanation tasks. We compare RoHeX with GNNExplainer and PGExplainer, two popular methods in this space. Results show that RoHeX achieves competitive or superior performance, highlighting the robustness of our variational denoising strategy even in homogeneous settings. In future work, we plan to extend RoHeX to more datasets.
>
> | BA-Shapes | GNNExplainer | PGExplainer | w/o VI | w/o He | RoHeX  |
> | --------- | ------------ | ----------- | ------ | ------ | ------ |
> | AUC (%)   | 0.8706       | 0.9172      | 0.9056 | 0.9213 | 0.9354 |
>
>
>
> [1] Lv, Q., Ding, M., Liu, Q., Chen, Y., Feng, W., He, S., ... & Tang, J. (2021, August). Are we really making much progress? revisiting, benchmarking and refining heterogeneous graph neural networks. In *Proceedings of the 27th ACM SIGKDD conference on knowledge discovery & data mining* (pp. 1150-1160).

---

> > ### Comment · Reviewer_KXBC · 2025-08-06
> > **Response to the Authors**
> >
> > Thank you for the detailed explanations for the concerns. After carefully reading the rebuttal, most of my concerns have been addressed, and I will keep my initial score at this moment.

---

> > > ### Author Response · Authors · 2025-08-06
> > >
> > > Thanks again for your insightful comments and suggestions which are very important to improve our paper. Since we have addressed the concerns and revised the manuscript accordingly, we kindly hope you might consider updating your score.

---

### Note · Authors · 2025-08-12

We believe that we have addressed all substantive concerns through detailed clarifications, additional experiments, and manuscript revisions. Three reviewers (KXBC, hbQA, 1WJ7) gave positive evaluations in quality, significance, and originality, recognizing the novelty of robust explainability for heterogeneous GNNs under structural noise, the soundness of our theoretical analysis, and the effectiveness of our denoising variational inference and relation-aware explanation framework. The key disagreements from Reviewer jCjz and our resolutions are:

- Uniform Degree Assumption in Theorem 3.1: The reviewer was concerned that this assumption was only mentioned in the proof. We have moved it into the theorem statement, clarified its scope, and empirically showed (JS divergence < 0.22) that it has negligible impact on performance.
- Derivation of Eq.21–23: The reviewer questioned the approximation steps. We expanded the detailed derivation, explained each approximation mathematically, and cited prior work showing this is standard in expectation analysis.
- Probability Bounds in Eq.3-5: The reviewer worried probabilities could exceed 1. We clarified that the probability is determined jointly by the noise budget and node degrees, and is within the range [0,1]. We added this explicitly to the manuscript.
- Definition of “Influence”: The reviewer found this undefined. We provide a rigorous definition.
- MAE/RMSE Evaluation Protocol: The reviewer noted mismatch between definitions and implementation. We clarified the metrics are computed on logits and revised the text accordingly.

In addition, we strengthened the paper with:

- We incorporated Accuracy Fidelity and Probability Fidelity from xPath, showing RoHeX achieves competitive results across multiple heterogeneous datasets.
- Additional benchmarks (IMDB, AMiner, BA-Shapes) demonstrating strong generalization;
- Clearer theoretical statements and definitions.

We note that despite these resolutions, Reviewer jCjz continued to raise the same points without acknowledging the changes. We trust the AC will weigh the majority consensus from other reviewers alongside these substantial clarifications and improvements.

In conclusion, RoHeX is the first systematic framework for robust heterogeneous GNN explanations under structural noise, with theoretical grounding, extensibility to other tasks, and consistently competitive results across diverse benchmarks. We believe it meets the standards for publication at NeurIPS.

---

### Decision · Program_Chairs · 2025-09-17

**Decision:**

Reject

**Comment:**

The paper proposes a GNN explanation model tailored to heterogeneous graphs. The model first addresses noise amplification through a denoising variational inference module, and then introduces a relation-aware explanation generator grounded in information-theoretic principles. The paper has merits that have been identified by the reviewers. These include parts of the methodology and its theoretical support. Nevertheless, the reviewers also pointed out limitations and concerns about the methodology.  Although some of these points have been thoroughly discussed during the discussion period, there are still points that need to be further addressed. Therefore, at this stage, I do not recommend acceptance. However, I encourage the authors to continue developing this line of work, considering that the core ideas proposed are promising.